# Beyond Loss Functions: Exploring Data-Centric Approaches with Diffusion Model for Domain Generalization

## Abstract

There has been a huge effort to tackle the Domain Generalization (DG) problem with a focus on developing new loss functions. Inspired by the image generation capabilities of the diffusion models, we pose a pivotal question: Can diffusion models function as data augmentation tools to address DG from a data-centric perspective, rather than relying on the loss functions? Our findings reveal that trivial cross-domain data augmentation (CDGA) along with the vanilla ERM using readily available diffusion models without additional finetuning outperforms state-of-the-art (SOTA) training algorithms.

This paper delves into the exploration of why and how this rudimentary data generation can outperform complicated DG algorithms. With the help of domain shift quantification tools, We empirically show that CDGA reduces the domain shift between domains. We empirically reveal connections between the loss landscape, adversarial robustness, and data generation, illustrating that CDGA reduces loss sharpness and improves robustness against adversarial shifts in data. Additionally, we discuss our intuitions that CDGA along with ERM can be considered as a way to replace the pointwise kernel estimates in ERM with new density estimates in the *vicinity of domain pairs* which can diminish the true data estimation error of ERM under domain shift scenario. These insights advocate for further investigation into the potential of data-centric approaches in DG.

## 1 Introduction

Out-of-distribution (OOD) generalization stands as a crucial capability for deep learning models in real-world scenarios. The prevalent setting for investigating OOD generalization is termed *domain generalization* (DG) Blanchard et al. (2011), involving multiple source domains to generalize to an unseen target domain. In DG problems, there is a shift between the training domains and the target domain which makes the models trained using Empirical Risk Minimization (ERM) Vapnik (1999b) struggle to maintain their performance in the target domain.

To enhance OOD generalization within the DG framework, researchers have proposed innovative loss functions—typically achieved by introducing regularizers to ERM—to facilitate the learning of domain-invariant mechanisms across domains. Nevertheless, none of these approaches consistently outperforms others across all datasets, as illustrated by results from the DomainBed benchmark (Gulrajani & Lopez-Paz, 2020). This observation suggests that a singular regularizer capable of capturing all invariances might not exist. We posit that the absence of such a universal regularizer arises from the diverse shifts present in each dataset, encompassing correlation shift, diversity shift, label shift, etc. Consequently, a rigid, data-independent regularizer may fall short in eliminating all types of spurious correlations and shifts. Additionally, the incorporation of sub-optimal regularizers can exacerbate optimization challenges, introducing excessive risk (Sener & Koltun, 2022), additional hyperparameters, and computational bottlenecks in ERM.

Rather than relying solely on traditional loss functions, recent advancements in generative foundation models open up new avenues for addressing the DG problem from a data-centric standpoint. Specifically, the capability of Denoising Diffusion Models (Ho et al., 2020; Song et al., 2020; Rombach et al., 2022) in generating high-fidelity synthetic images offers an innovative approach for advanced data augmentation,

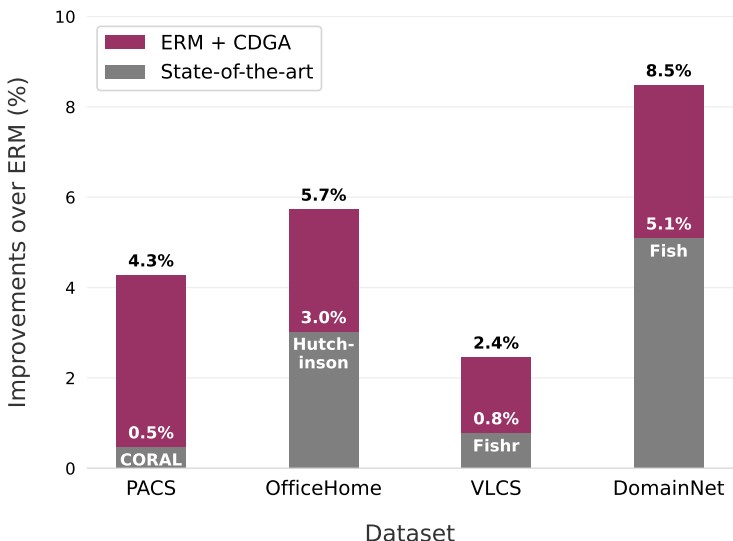

Figure 1: Improvements of CDGA + ERM and state-of-the-art (SOTA) DG training methods over ERM across four dataset using DomainBed benchmark. See Section 4 for experimental details. CDGA + ERM outperforms SOTA algorithms across all datasets, highlighting the absence of a singular algorithm achieving best results across datasets.

enabling the creation of domain-invariant images to enhance OOD generalization. To examine this hypothesis, we employ a straightforward *Cross Domain Generative Augmentation* (CDGA) method. In CDGA, synthetic images are generated conditioned on images or text descriptions from all possible combinations of the training domains using a pre-trained latent diffusion model (LDM) (Rombach et al., 2022). In Figure 1, we show that applying vanilla ERM to combined generated and real images outperforms the previous state-of-the-art algorithms *across all datasets.*

This paper delves into an exploration of the reasons and mechanisms behind the superior performance of CDGA's simplistic data generation strategy compared to complex DG training algorithms. Our investigation involves quantifying and visualizing domain shifts across domains of generated synthetic images, validating that cross-domain data generation mitigates the gap between domains. We also discuss our intuitions that CDGA can be seen as a way to replace pointwise kernel estimates in ERM with new density estimates in the proximity of *domain pairs.* This modification to ERM can the inherent data estimation error in the presence of domain shift, subsequently enhancing its out-of-distribution (OOD) performance. Furthermore, our empirical results establish connections between the loss landscape, adversarial robustness, and data generation, revealing that cross-domain data generation lessens loss sharpness and improves robustness against adversarial shifts in data. To the best of our knowledge, we are the first to utilize latent diffusion models as a data-centric approach for DG.

Our primary contributions are as follows:

1. Demonstrating superior performance, our study reveals that combining CDGA with vanilla ERM outperforms state-of-the-art DG training algorithms. We validate this across four datasets using three model selection strategies.

2. Employing various metrics such as transfer measure, diversity shift, and near-duplicate quantification, we empirically demonstrate that CDGA reduces distribution shift among domains, attributing to its superior performance.

3. Providing possible intuitions that CDGA can be seen as a way to replace pointwise kernel estimates in ERM with new density estimates in the proximity of *domain pairs* similar to the Vicinal Risk Minimization principle (VRM) (Chapelle et al., 2000) within DG setup.

4. Through empirical analysis, we calculate the loss landscape sharpness of CDGA during training with ERM, showcasing its reduced sharpness compared to vanilla ERM. Additionally, we demonstrate the robustness of CDGA + ERM against two adversarial shift attacks.

5. Our extensive ablation studies compare single-domain and cross-domain data generation, underscoring the necessity of cross-domain generation for substantial improvements. Furthermore, our ablation on the size of generated data highlights its potential as a method for mitigating class imbalance challenges.

## 2  Problem Settings and Related Work

We denote our prediction model as $f$ and its parameters as $\boldsymbol{\theta}$. In DG, the goal is to learn a shared model from $n$ source (train) population domains (environments) $\{\mathcal{E}_1, \ldots, \mathcal{E}_n\}$, to generalize to an unseen target domain $\mathcal{T}$. For a given domain $\mathcal{E}$, the classification loss is:

$$\mathcal{L}_{\mathcal{E}}(\theta) = \mathbb{E}_{(x,y)\sim\mathcal{E}}[\ell(f(x;\theta), y)], \tag{1}$$

where each $x$ and $y$ are the input data point and its corresponding label and $\ell(f(x;\theta), y)$ is the cross entropy loss between $f(x;\theta)$ and $y$.

**ERM** The standard baseline for training deep learning models is ERM Vapnik (1999b), which minimizes the average of losses over the entire available training domains i.e., $\{S_1, \ldots, S_n\}$

$$\min_{\theta} \sum_i \frac{1}{|S_i|} \sum_{k=1}^{|S_i|} \ell(f(x_k^i; \theta), y_k^i) \tag{2}$$

where $(x_k^i, y_k^i) \sim \mathcal{E}_i$ is an i.i.d. sample from the training set $S_i$ of domain $i$ with $|S_i|$ samples, $x_k^i$ is the $k$-th data point in $S_i$ and $y_k^i$ is its associated label. However, in the case of domain shift between domains, the pointwise kernel estimation of true data distribution proposed by ERM in Eq. 2, becomes less accurate which results in a lack of OOD generalization of ERM.

**Improving the Loss Function:** In the pursuit of improving OOD performance within ERM, diverse avenues have been explored. Notable efforts include robust optimization (Sagawa et al., 2020; Hu et al., 2018), invariant representation learning on the feature level (Sun & Saenko, 2016; Ganin et al., 2016; Li et al., 2018; Tzeng et al., 2014), and classifier head adjustments (Arjovsky et al., 2019). Additionally, advancements in loss functions, such as those discussed by Krueger et al. (2021), reveal their efficacy in enhancing domain generalization. Further insights stem from investigations into loss gradients/Hessians (Parascandolo et al., 2020; Shahtalebi et al., 2021; Koyama & Yamaguchi, 2020; Shi et al., 2021; Hemati et al., 2023), showcasing the multifaceted approaches taken to fortify ERM against domain shifts.

**Data Augmentation for DG:** Various strategies enhance OOD performance in Empirical Risk Minimization (ERM). Classic data augmentation, as explored by Gulrajani & Lopez-Paz (2020), demonstrates improved results under the DomainBed evaluation protocol. Ilse et al. (2021) introduces Select Data Augmentation, a method that identifies transformations detrimentally affecting validation accuracy. Mixup (Zhang et al., 2017) generates mixed data points and soft labels through linear convex combinations from two classes, while MixStyle (Zhou et al., 2021) combines per-sample feature statistics (mean and variance) across domains. Somavarapu et al. (2020) introduces a stylization transformation based on in-domain data. To the best of our knowledge, we are the first to utilize latent diffusion models as a data-centric approach for DG.

**Denoising diffusion models and their applications.** Recent advances in diffusion-based generative models (Ho et al., 2020; Song et al., 2020; Rombach et al., 2022; Zhang & Agrawala, 2023) demonstrate their capability to achieve SOTA image quality. Additionally, works such as Unclip (Ramesh et al., 2022) have successfully integrated Foundation models like CLIP (Radford et al., 2021) with stable diffusion, introducing new generative functionalities such as image-to-image, text-to-image, image variation, and image mixing to diffusion-based models. The application of diffusion models in representation learning has also been explored, as exemplified by StableRep proposed by Tian et al. (2023). In a setup akin to SimCLR (Chen et al., 2020), they demonstrated that synthetic images generated by stable diffusion models can enhance self-supervised learning.

# 3 Cross Domain Generative Augmentation

In this section, we provide a detailed description of CDGA. CDGA utilizes LDM to perform a transformation denoted by $\mathcal{M}(\cdot)$. This transformation takes two arguments: a data point in one domain and a guidance attribute in another domain from the same class. Formally,

$$\widetilde{x}_k^{i,j} = \mathcal{M}(x_k^i, \texttt{guide}^j), \tag{3}$$

where $\widetilde{x}_k^{i,j}$ is a synthetic image transformed from domains $i$ and $j$, generated from the $k$-th sample in $S_i$. The attribute $\texttt{guide}^j$ serves as guidance towards another domain, $S_j$, within the same class. The workflow is illustrated in Figure 2),

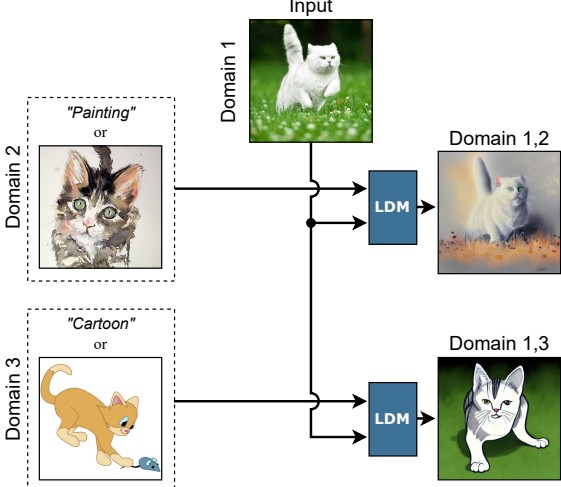

Figure 2: Illustration of CDGA. For each input image of a domain, we generate a new image using the image or the description of another domain.

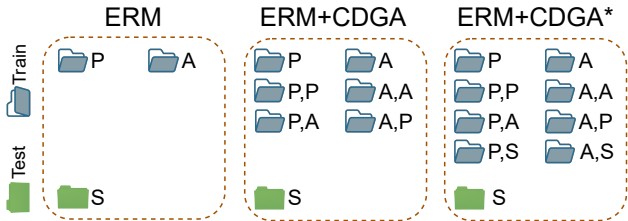

Figure 3: Illustration of the implementation structure of ERM, CDGA, and CDGA* on PACS dataset when using P and A domains as training and S as target domain.

In CDGA, each data point in domain $S_i$ undergoes transformation to all $n$ domains, including its own domain. This augmentation increases the number of samples for domain $S_i$ from $|S_i|$ to $(b \times n + 1) \times |S_i|$, where $n$ is the number of training domains, $|S_i|$ is the number of data points in $S_i$, and $b$ is the generation batch size. Furthermore, we introduce CDGA*, where we assume access to a guidance attribute of the target domain. In this scenario, the size of domain $S_i$ increases from $|S_i|$ to $(b \times (n + 1) + 1) \times |S_i|$. For implementing CDGA, we use offline augmentation where we first generate images between each pair of training domains and then start the training process. As an example, the folder structure of our implementation for the PACS dataset when using P and A domains as train domains and S domain for test domain is illustrated in Figure 3. For all the methods, we set generation batch size $b = 1$ unless stated otherwise.

**CDGA with Prompt Guidance (CDGA-PG)**: In CDGA-PG, given the $k$-th image in $S_i$, i.e., $x_i^k$, the guidance attribute $\texttt{guide}^j$ is a domain description text prompt that represents the same class in $S_j$. Having the image and the prompt guidance, we use the LDM to generate $b$ synthetic images which we expect to

interpolate domains $i$ and $j$ for the same class. For each image in $S_i$, we perform these image translations for all the training domains $j$, $\forall j \in \{1, ..., n\}$. We also consider the scenario where we can utilize the target domain description, i.e., $\texttt{guide}^{\mathcal{T}}$ as the guidance.

**CDGA with Image Guidance (CDGA-IG)**: For scenarios where a text prompt description of domains is not available, CDGA-IG is used where the guidance is an image from $S_j$ instead of a text description. More precisely, in CDGA-IG we attempt to mix two images from two different domains which is also known as the image mixer in the literature.

## 4 CDGA Outperforms SOTA

In this section, we compare CDGA + ERM with SOTA DG training methods, demonstrating its superior performance. We assess CDGA and CDGA* on for datasets, namely VLCS (Fang et al., 2013), PACS (Li et al., 2017), OfficeHome (Venkateswara et al., 2017), and DomainNet (Peng et al., 2019), using the DomainBed benchmark (Gulrajani & Lopez-Paz, 2020). This benchmark has gained popularity as a fair and standard evaluation platform for domain generalization algorithms. The evaluation process involves comparing DG algorithms across 20 hyperparameter choices and 3 trials, utilizing three distinct model selection techniques. To demonstrate CDGA's effectiveness, we present its evaluation results using the DomainBed benchmark in Tables 1-4. The tables follow a format of presenting the **first** and second results. For brevity, we report only the top five performing algorithms for each model selection, with full results available in the appendix. Examining Tables 1-4, CDGA* consistently achieves SOTA performance across all datasets and model selection techniques. Specifically, we applied prompt guidance for PACS, OfficeHome, and DomainNet, while using image guidance (i.e., image mixer) for VLCS. The code implementation for deploying CDGA-generated data within the DomainBed scheme is detailed in Appendix H.

Table 1: DomainBed benchmark for **training-domain validation set** model selection method.

| Algorithm | PACS | OfficeHome | DomainNet | Avg |
|---|---|---|---|---|
| ERM | $85.5 \pm 0.2$ | $66.5 \pm 0.3$ | $40.9 \pm 1.8$ | 64.3 |
| CORAL | $86.2 \pm 0.3$ | $\underline{68.7} \pm 0.3$ | $41.5 \pm 0.1$ | 65.5 |
| SagNet | $86.3 \pm 0.2$ | $68.1 \pm 0.1$ | $40.3 \pm 0.1$ | 64.9 |
| Fish | $85.5 \pm 0.3$ | $68.6 \pm 0.4$ | $42.7 \pm 0.2$ | 65.6 |
| Fishr | $85.5 \pm 0.4$ | $67.8 \pm 0.1$ | $41.7 \pm 0.0$ | 65.0 |
| HGP | $84.7 \pm 0.0$ | $68.2 \pm 0.0$ | $41.1 \pm 0.0$ | 64.7 |
| ERM + CDGA-PG | $\underline{88.5} \pm 0.5$ | $68.2 \pm 0.6$ | $\underline{43.7} \pm 0.1$ | $\underline{66.6}$ |
| ERM + CDGA*-PG | $\mathbf{89.5} \pm 0.3$ | $\mathbf{70.8} \pm 0.6$ | $\mathbf{44.8} \pm 0.0$ | $\mathbf{68.4}$ |

Table 2: DomainBed benchmark for **leave-one-domain-out cross-validation** model selection.

| Algorithm | PACS | OfficeHome | DomainNet | Avg |
|---|---|---|---|---|
| ERM | $83.0 \pm 0.7$ | $65.7 \pm 0.5$ | $40.6 \pm 0.2$ | 63.1 |
| CORAL | $82.6 \pm 0.5$ | $68.5 \pm 0.2$ | $41.1 \pm 0.1$ | 64.1 |
| SagNet | $82.3 \pm 0.1$ | $67.6 \pm 0.3$ | $40.2 \pm 0.2$ | 63.4 |
| MLDG | $82.9 \pm 1.7$ | $66.1 \pm 0.5$ | $41.0 \pm 0.2$ | 63.3 |
| HGP | $82.2 \pm 0.0$ | $67.5 \pm 0.0$ | $41.1 \pm 0.0$ | 63.6 |
| Hutchinson | $84.8 \pm 0.0$ | $68.5 \pm 0.0$ | $41.4 \pm 0.0$ | 64.9 |
| ERM + CDGA-PG | $\underline{86.8} \pm 0.4$ | $\underline{68.7} \pm 0.4$ | $\underline{43.6} \pm 0.1$ | $\underline{66.2}$ |
| ERM + CDGA*-PG | $\mathbf{88.4} \pm 0.5$ | $\mathbf{70.2} \pm 0.4$ | $\mathbf{44.8} \pm 0.0$ | $\mathbf{67.8}$ |

## 5 CDGA Reduces Domain Shift

In this section, we empirically confirm that CDGA reduces domain shift. To validate the efficacy of CDGA in mitigating domain shift, we employ five domain shift quantification techniques from the literature on the

Table 3: DomainBed benchmark **test-domain validation set (oracle)**model selection method.

| Algorithm | PACS | OfficeHome | DomainNet | Avg |
|---|---|---|---|---|
| ERM | 86.7 ± 0.3 | 66.4 ± 0.5 | 41.3 ± 0.1 | 64.8 |
| Mixup | 86.8 ± 0.3 | 68.0 ± 0.2 | 39.6 ± 0.1 | 64.8 |
| MLDG | 86.8 ± 0.4 | 66.6 ± 0.3 | 41.6 ± 0.1 | 65.0 |
| CORAL | 87.1 ± 0.5 | 68.4 ± 0.2 | 41.8 ± 0.1 | 65.8 |
| SagNet | 86.4 ± 0.4 | 67.5 ± 0.2 | 40.8 ± 0.2 | 64.9 |
| Fish | 85.8 ± 0.6 | 66.0 ± 2.9 | 43.4 ± 0.3 | 65.1 |
| Fishr | 86.9 ± 0.2 | 68.2 ± 0.2 | 41.8 ± 0.2 | 65.6 |
| Hutchinson | 86.3 ± 0.0 | 68.4 ± 0.0 | 41.9 ± 0.0 | 65.5 |
| ERM + CDGA-PG | 89.6 ± 0.3 | 68.8 ± 0.3 | 44.4 ±0.1 | 67.2 |
| ERM + CDGA*-PG | **90.4** ± 0.3 | **70.2** ± 0.2 | **44.8** ±0.0 | **68.5** |

Table 4: DomainBed benchmark on **VLCS** dataset.

| Method | Training domain | Leave-one-domain-out | Oracle |
|---|---|---|---|
| ERM | 77.5 ± 0.4 | 77.2 ± 0.4 | 77.6 ± 0.3 |
| CORAL | 78.8 ± 0.6 | 78.7 ± 0.4 | 77.7 ± 0.2 |
| SagNet | 77.8 ± 0.5 | 77.5 ± 0.3 | 77.6 ± 0.1 |
| Fishr | 77.8 ± 0.1 | 78.2 ± 0.0 | 78.2 ± 0.2 |
| HGP | 77.6 ± 0.0 | 76.7 ± 0.0 | 77.3 ± 0.0 |
| Hutchinson | 76.8 ± 0.0 | **79.3** ± 0.0 | 77.9 ± 0.0 |
| ERM + CDGA-IG | **78.9** ± 0.3 | 77.9 ± 0.5 | **79.5** ± 0.1 |

PACS dataset. Specifically, we utilize t-SNE visualization of feature embeddings, near-duplicate analysis (Oquab et al., 2023), transferability (Zhang et al., 2021; Hemati et al., 2023), and diversity shift metrics (Ye et al., 2022) to quantify the shift between domains.

## 5.1 Domain shift Visualization

To visualize domain shifts in CDGA-based data for the class "dog" across all domains (P, A, C, and S), we generate synthetic images for A → A, A → P, A → C, and A → S. Subsequently, we utilize the pretrained CLIP ViT-B/32 image encoder (Radford et al., 2021) to extract features from both real and synthetic images. These features are then projected onto a two-dimensional space using t-SNE and presented in Figure 4. Notably, the cross-domain synthetic images effectively interpolate between different domains, addressing the desired distribution shift. In Figure 4, examining domains A (in red) and S (in pink) reveals a significant distribution shift in their two-dimensional representations, despite all images belonging to the dog class. However, A → S synthetic images seamlessly bridge the gap between A and S representations. Refer to Figure 15 in the appendix for t-SNE plots of other classes.

## 5.2 Transferability Measurement

Transferability is another recent approach proposed by Zhang et al. (2021) to quantify the domain shift. The transferability measure is an upper bound of the difference between the source and the target domain excess risks. Subsequently, Hemati et al. (2023) showed that an upper bound for transferability measure is $\frac{1}{2}\delta^2\|\boldsymbol{H}_{\mathcal{T}} - \boldsymbol{H}_{\mathcal{S}}\|_2 + o(\delta^2)$ where $\delta$ is a constant, $\boldsymbol{H}_{\mathcal{T}}$ and $\boldsymbol{H}_{\mathcal{S}}$ are target and source classifier head's Hessians. Following these findings, to quantify the dynamics of domain shift, we monitor classifier heads Hessian distances between all possible domain pairs through the training steps for ERM and CDGA, where we set domains P, A, and C as train (source) and domain S as target. Figure 5 shows the difference between the classifier head's Hessians given data from domains A and S during the steps. We see that CDGA and CDGA* lead to lower classifier head's Hessian difference and subsequently smaller transferability and domain shift

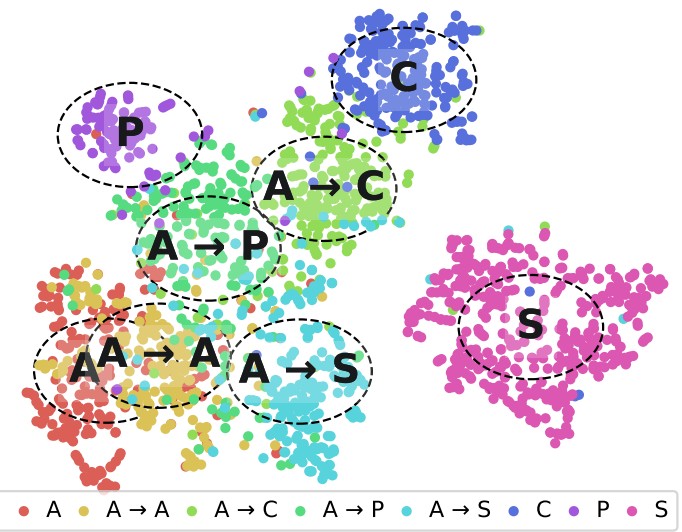

Figure 4: The *t*-SNE plot of features extracted from the original PACS dataset and generated images by CDGA from A domain. This figure shows that CDGA can fill the gap between the original domains. Check Section 5 for details.

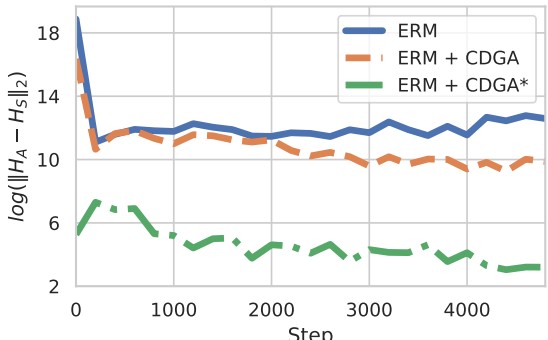

Figure 5: Classifier head Hessian difference between domains P & S during training.

compared to ERM. This pattern consistently exists for other domain pairs where the full results are presented in Figure 14 in the appendix.

### 5.3 Diversity Shift

Ye et al. (2022) proposed a numerical method to measure diversity shift which is equivalent to total variation (Zhang et al., 2021) to quantify domain shift. Diversity shift is usually due to the novel domain-specific features in the data. We employ the proposed algorithm by Ye et al. (2022) to quantify and compare diversity shift between training domains and the target domain in a leave-one domain out scheme for PACS real data, CDGA-PACA, and CDGA*-PACS datasets. Figure 8 shows both CDGA and CDGA* reduce the diversity shift between training domains and the target domain.

### 5.4 Near-duplicate Analysis

We employ near-duplicate image detection on images generated using CDGA to quantify the similarity between the generated and original images in each domain. Following the self-supervised image retrieval technique outlined in (Oquab et al., 2023), we utilize the pretrained CLIP ViT-B/32 image encoder (Radford et al., 2021) to extract embeddings and calculate cosine similarity between original and generated images. For each original image, if at least one image in a generated domain exhibits a cosine similarity above 0.95,

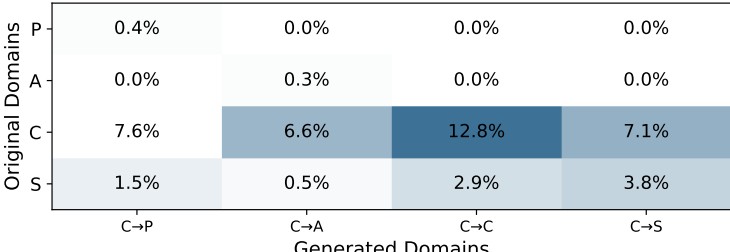

Figure 6: Heat map of the percentage of near-duplicates of each original domain in the generated domains. This table shows that using target-domain description results in more near-duplicate images.

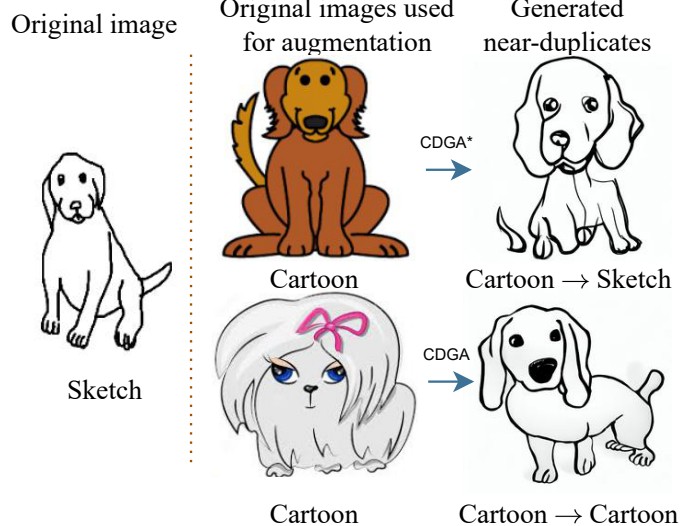

Figure 7: Examples of near-duplicates (right-most column) found for the dog image in Sketch domain (left-most column) that are generated using CDGA from the original images (middle column).

we categorize the original domain as having a near-duplicate. Figure 6 provides a summarized view of this experiment for the case of generated images from domain C, while the complete results are available in Figure 12 in the appendix. In Figure 6, we report, for each original domain, the percentage of near-duplicates relative to the original domain size. Clearly, generating synthetic images within the manifold between training domains allows us to obtain examples that are near-duplicates of the target domain. Figure 7 showcases some of the near-duplicates identified for a sample image in the S domain using this technique. Additional examples can be found in Figure 13 in the appendix.

## 6  Intuitive Discussion: CDGA with ERM an approximate Extention of Vicinal Risk Minimization Principle to DG Setup

In this section, we attempt to provide an intuitive justification for the reasons behind the success of CDGA. Our justification relies on the Vicinal Risk Minimization principle (VRM) (Chapelle et al., 2000) initially introduced by Vapnik (1999a). The motivation behind VRM is to improve true data distribution estimation in ERM. First note that we can rewrite ERM (Vapnik, 1999a) loss in Eq. 2 as

$$\min_{\boldsymbol{\theta}} \sum_i \frac{1}{|S_i|} \sum_k^{|S_i|} \int \ell(f(x;\theta),y)\delta(x;x_k^i)\delta(y;y_k^i)dxdy,$$

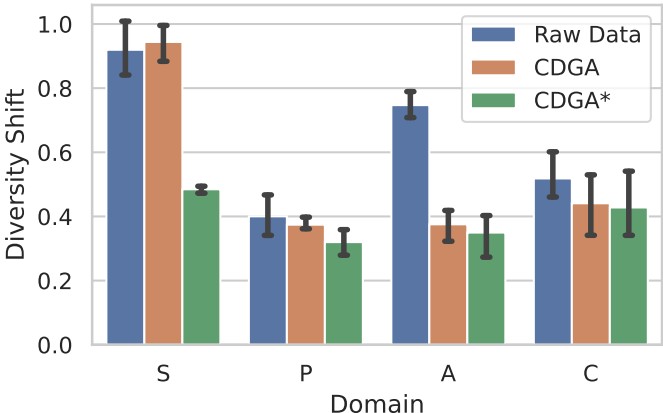

Figure 8: Diversity shift as a measure of iid-ness of PACS (raw data) against CDGA, and CDGA* augmented datasets. Each column is the target domain and the rest of the domains are training domains. CDGA and CDGA* reduce the diversity shift.

where $\delta(\boldsymbol{x}; x_k^i)$ is the Dirac delta distribution for the $k$-th data point in $S_i$. To unify this formulation with the loss on true distribution data we can rewrite Eq. 2 as

$$\min_{\theta} \mathbb{E}_{i \sim \text{Unif(n)}} \mathbb{E}_{(x,y) \sim \widehat{\mathcal{E}}_i} [\ell(f(x; \theta), y)], \qquad (4)$$

where Unif(n) is the discrete uniform distribution over 0 to $n$ and the sample distribution $\widehat{\mathcal{E}}_i$ is defined as

$$\widehat{\mathcal{E}}_i(x) = \frac{1}{|S_i|} \sum_k^{|S_i|} \delta(x; x_k^i) \delta(y; y_k^i). \qquad (5)$$

To improve the true data estimation error of ERM, this is necessary to improve the sample distribution $\widehat{\mathcal{E}}_i$. To this end, the VRM principle suggests replacing the point-wise estimate in ERM i.e., $\delta(x; x_k^i)$ in eq. 5 with some better kernel estimate of the density in the vicinity of the data point $k$ within $S_i$ which we call $K(x; x_k^i)$. An example for $K(x; x_k^i)$ can be Gaussian kernel functions that act as smooth $\delta(x; x_k^i)$ functions. In this case, the VRM sample distribution represented by $\widetilde{\mathcal{E}}_i$ is written as

$$\widetilde{\mathcal{E}}_i(x) = \frac{1}{|S_i|} \sum_k^{|S_i|} K(x; x_k^i) \delta(y; y_k^i), \qquad (6)$$

and the overall loss is obtained by replacing $\widehat{\mathcal{E}}_i(x)$ in Equation 4 by $\widetilde{\mathcal{E}}_i(x)$. In practice, VRM is implemented by employing data augmentation along with ERM. VRM suggests that data augmentation can improve ERM estimation error by adding additional synthetic examples from the vicinity distribution around each observation *within each domain* in the data. We argue in the DG setting, if we only employ classic data augmentation the estimation error of VRM is still high. This is because, in the DG setting, the estimation error of true data distribution by ERM is mainly caused by the distribution shift *between domains* which cannot be fully addressed by simple data augmentation techniques within each domain.

Extending the VRM (Chapelle et al., 2000) principle to the DG setup, a possible intuitive justification on the success of CDGA is that employing CDGA replaces the pointwise estimates in ERM with new density estimates that can be in the *vicinity of domain pairs* so that the distribution shift between domains is further reduced. We believe this step can potentially reduce the estimation error induced by ERM under the domain shift scenario. To see the difference between classic augmentation and CDGA more clearly, first, we need to define the projection operator:

**Definition 6.1 (domain projection operator).** Given a metric $d$ defined on $\text{supp}(\mathcal{E}_i) \cup \text{supp}(\mathcal{E}_j)$, the domain projection operator $P_{i \to j}(\cdot)$ that projects a data point $x$ in $\mathcal{E}_i$ (i.e., $x \in \text{supp}(\mathcal{E}_i)$) onto domain $j$ is defined as

$$P_{i \to j}(x) = \underset{z \in \text{supp}(\mathcal{E}_j)}{\text{argmin}} \ d(z, x).$$

The projection operator gives the mathematical definition for what we mean by "finding the closest point on another domain." For example, if $x$ is a cat image of painting style ($\mathcal{E}_i$), then $P_{i \to j}(x)$ would mean the most similar image to $x$ in the sketch domain $\mathcal{E}_j$. The projection operator relies on the metric $d$, which measures the similarity of two samples. This similarity can be implemented, e.g., by computing the $L_2$ distance in some latent feature embedding space.

Define the smoothed sample distribution of $\widetilde{\mathcal{E}}_i$ projected to domain $j$ by $\widetilde{\mathcal{E}}_{i,j}$:

$$\widetilde{\mathcal{E}}_{i,j}(x) = \frac{1}{|S_i|} \sum_k^{|S_i|} K(x; P_{i \to j}(x_k^i)) \delta(y; y_k^i). \tag{7}$$

where $i$ and $j$ are domain counters and $k$ is the data point index, $K(x; P_j(x_k^i))$ represents the vicinity of $k$-th data point in $\mathcal{E}_i$ projected onto $\mathcal{E}_j$ which is label invariant. Similar to VRM, the expression for the final loss function can be obtained by replacing $\widehat{\mathcal{E}}_i(x)$ with $\frac{1}{n} \sum_j \widetilde{\mathcal{E}}_{i,j}(x)$ in equation 4.

In practice, thanks to image manipulation of diffusion models, such projection to other domains and sampling from the vicinity of projected data points has become feasible (approximately). To this end, in CDGA, we employ LDM, denoted by $\mathcal{M}(\cdot)$ which takes two arguments, one is a data point in a domain and the second argument is a guidance attribute in another domain from the same class i.e., $\widetilde{x}_k^{i,j} = \mathcal{M}(x_k^i, \texttt{guide}^j) \sim P(\cdot)$ where $P(\cdot)$ is defined $P(\cdot) = \frac{K(\cdot; P_{i \to j}(x_k^i))}{\int K(\cdot; P_{i \to j}(x_k^i))\, dx}$. Effectively, we are drawing an augmented sample near the projected sample in the target domain,[1] and the kernel estimation relies on the LDM we choose. In this case the proposed proposed loss realized by CDGA along with ERM has the following form.

$$\min_\theta \mathbb{E}_{i \sim \text{Unif(n)}} \mathbb{E}_{j \sim \text{Unif(n)}} \mathbb{E}_{(x,y) \sim \widetilde{\mathcal{E}}_{i,j}} [\ell(f(x; \theta), y)], \tag{8}$$

Since our kernel relies on the LDM we use, the estimation error comparison between equation 8 and equation 5 seems infeasible. More theoretical exploration along this direction is left as future work.

## 7 Ablation Studies

### 7.1 Mitigating Class Imbalance

CDGA can also be utilized to mitigate the class imbalance problem in datasets where the number of instances in each class of each domain is not equal. In such scenarios, one can use a different $b$ for each class of the data such that after generating samples, the number of instances in each class of generated domains becomes equal. We test the effectiveness of CDGA method in balancing the OfficeHome dataset (which is highly imbalanced) through the DomainBed benchmark. More specifically, for every class $c$ and domain $S_j$, we find the number of samples $n(S_j, c)$ and then we find $m = \max_{c,j} n(S_j, c)$ which is 100 for OfficeHome. Then for every domain $S_j$ and class $c$ we set $b = \frac{m}{n(S_j, c)}$ which leads to larger batch size for domains and classes with fewer data points and subsequently balances the dataset. The results of this experiment are presented in Table 5. Clearly, by choosing $b$ in a way that the dataset is more balanced, the OOD generalization has been further improved.

Table 5: OOD accuracy of models with and without balanced generation in OfficeHome dataset .

| Method | Training domain | Leave-one -domain-out | Oracle |
|---|---|---|---|
| ERM | $66.5 \pm 0.3$ | $65.7 \pm 0.5$ | $66.4 \pm 0.5$ |
| ERM + CDGA ($b = 1$) | $\underline{68.2} \pm 0.6$ | $\underline{68.7} \pm 0.4$ | $\underline{68.6} \pm 0.3$ |
| ERM + CDGA ($b = \frac{m}{n(\mathcal{E}_j, c)}$) | $\mathbf{69.9} \pm 0.2$ | $\mathbf{69.7} \pm 0.4$ | $\mathbf{70.0} \pm 0.7$ |

---

[1] We implicitly assume that $P_{i \to j}(x)$ is a singleton for any $x$. Otherwise, we can extend the definition of $\widetilde{\mathcal{E}}_{i,j}$ by averaging over all "closest" projections.

## 7.2 Single Domain Generative Augmentation (SDGA) vs CDGA

To show the advantage of employing cross-domain data to mitigate domain shift, We also explore the SDGA method, where, unlike CDGA, the image from $S_i$ is augmented only from the guidance of the same domain i.e., $\mathcal{M}(\texttt{guide}^i)$, where guidance can either be an image (SDGA-IG) or a prompt (SDGA-PG). In SDGA-PG, for each image in $S_i$, i.e., $x_k^i$ we create prompt guidance $\texttt{guide}^i$ that can contain label and/or domain information from $S_j$ and feed $\texttt{guide}^i$ to the LDM. In SDGA-IG, for each image from $S_i$, i.e., $x_k^i$ we construct guidance that contains both $x_k^i$ and label information. To compare variations of CDGA and SDGA, we evaluate them on the PACS dataset using the DomainBed benchmark with twenty different hyperparameters and one trial. The results of this experiment are presented in Table 6. Clearly, CDGA consistently outperforms all variations of SDGA. As some suggestions for use cases, we believe as long as the objective is maximum OOD performance, either textual or visual descriptions of different domains are accessible, and there is no computational bottleneck, the CDGA is a better choice compared with SDGA. On the other hand, if we do not have access to the domain descriptions, or we aim to achieve OOD improvement as fast as possible with fewer training examples, SDGA can be a better option.

Table 6: OOD accuracy of models trained with variations of CDGA and SDGA on PACS dataset using Domainbed benchmark.

| Method | Training domain | Leave-one -domain-out | Oracle |
|---|---|---|---|
| ERM | $85.5 \pm 0.2$ | $83.0 \pm 0.7$ | $86.7 \pm 0.3$ |
| + SDGA-PG (label) | $86.1 \pm 0.5$ | $83.7 \pm 1.0$ | $87.3 \pm 0.5$ |
| + SDGA-PG (label+domain) | $85.9 \pm 1.0$ | $84.6 \pm 0.8$ | $87.5 \pm 1.1$ |
| + SDGA-IG (label) | $87.5 \pm 0.6$ | $86.5 \pm 1.1$ | $89.5 \pm 0.3$ |
| + CDGA-PG (canny edge) | $86.8 \pm 0.9$ | $79.2 \pm 3.6$ | $87.8 \pm 0.3$ |
| + CDGA-PG | $\underline{88.5} \pm 0.5$ | $\underline{86.8} \pm 0.4$ | $\underline{89.6} \pm 0.3$ |
| + CDGA*-PG | $\mathbf{89.5} \pm 0.3$ | $\mathbf{88.4} \pm 0.5$ | $\mathbf{90.4} \pm 0.3$ |

## 7.3 Scaling Law of Data Size

In this experiment, we aim to measure the effect of generation batch size $b$ which determines the final augmented dataset size on the OOD generalization of models trained with CDGA. To this end, on the PACS dataset, using DomainBed benchmark with 20 choices of hyperparameters and 1 trial, we apply CDGA-PG with $b$ equal to 1, 2, 3, and 4. We conduct this experiment for two models: ResNet-18 model pretrained on ImageNet and ResNet-18 model with random initialization. As we see from Table 7, for both pretrained and random initializations, increasing $b$, i.e., increasing the data size, further improves OOD generalization. However, for the pretrained model, the performance gets saturated which can be due to the capacity of the model.

# 8 Does superior performance of the CDGA method comes from cross-domain transfer or the improvement is just because of extra synthetic images from a pre-trained diffusion model

A critical question that may be raised is whether the benefit of the CDGA method comes from cross-domain transfer and not from generating extra synthetic data from a pre-trained diffusion model. We do believe the superior OOD performance of CDGA is due to cross-domain transfer which removes the distribution shift between domains and not just because we are using a model that is pre-trained and generates synthetic data. Here we provide our arguments to support this claim.

- **The reason behind beter OOD generalization is synthetic samples *between domains (cross domain augmentation)* and not just additional samples from a pre-trained diffusion**

**model**. By looking more closely at Table. 6, we observe that although in both SDGA and CDGA techniques the same pre-trained diffusion model has been used, the OOD generalization improvement for CDGA is much higher than the improvement by SDGA. This observation suggests that the superior OOD generalization is not just due to incorporating additional syntethic samples. Instead, this improvement is due to employing **additional syntethic samples that reduce the distribution shift across domains**. The claim that CDGA actually reduces the distribution shift more than SDGA is experimentally supported in Figure 4 where for example A $\rightarrow$ C synthetic samples from CDGA are filling the gap between domains A and C while for SDGA, only A $\rightarrow$ A samples are generated which are not able to reduce the domain shift between domain A and C. More examples on this can be found in Figure 15. In other words, while both SDGA and CDGA employ the same pre-trained model, CDGA is able to generate samples between domains and reduce the distribution shift better than SDGA and as a result achieve better OOD generalization.

- **We are already empoying pre-trained ResNet for the domainbed experiments and it fails to provide good OOD generalization**. Consider the experimental results from the domainbed benchmark where a large imageNet pre-trained model (ResNet) is used for all algorithms in Tables 1-4 in our paper. As you can see, even with a large pre-trained model, the model fails to achieve good OOD generalization while employing CDGA further boosts the performance.

These arguments suggests that merely employing a pre-trained model, no matter this is for training or data augmentation, does not necessarily improve the OOD generalization. In fact, the contributing factor is cross domain ability of CDGA which reduces the domain shift between domains.

Table 7: Effect of generation batch size $b$ on CDGA for PACS dataset with different initialization.

| Initialization | $b$ | Training domain | Leave-one -domain-out | Oracle |
|---|---|---|---|---|
| Random | 1 | 66.1 | 61.6 | 65.0 |
| | 2 | 70.2 | 69.2 | 70.3 |
| | 3 | 71.8 | 71.8 | 74.5 |
| | 4 | **73.9** | **73.2** | **74.6** |
| Pre-trained | 1 | 87.0 | 86.4 | 89.0 |
| | 2 | 87.4 | 86.2 | 89.0 |
| | 3 | **88.4** | **89.1** | 88.9 |
| | 4 | **88.4** | 88.2 | **89.2** |

## 9    Conclusions

In this paper, we showed that a simple cross domain generative augmentation (i.e., CDGA) alongside ERM surpasses SOTA DG algorithms in the standard DomainBed benchmark. Additionally, empirical results show by employing various distribution shift quantification methods, we observe a significant reduction in distribution shift between training domains after applying CDGA. Furthermore, we conduct comprehensive ablation studies, particularly focusing on adversarial robustness, loss landscape analysis, and data scaling laws. Notably, the use of CDGA enhances adversarial robustness and reduces the sharpness of the loss landscape, both contributing to improved model generalization. Finally, intuitively, we establish that CDGA with ERM approximately could be considered as an extension of the VRM principle to the DG setup. Our work provides a novel data-centric point of view for domain generalization, in the era when AI Generated Content (AIGC) becomes more and more popular.

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
