# A    Implementation Details

**Models**: We use the pretrained, version 1.4 of stable diffusion (Rombach et al., 2022) without finetuning as our base LDM. For the implementation of CDGA-IG, we use the image mixer that has been fine-tuned by Justin Pinkney at Lambda Labs (Pinkney, 2023) to accept CLIP image embeddings. For image generation, we do not tune any hyperparameters (e.g., strength, steps, etc) and all the parameters are set to their default values of the (Rombach et al., 2022) repository.

**Prompts**: For CDGA-PG, we use both the classes of the images and the domain description in the text prompts as guidance. The complete list of the prompts used for each domain in each dataset is in appendix G.

**Hardware**: We use two clusters of four V100 NVIDIA GPUs for generation and benchmarks.

## A.1    Loss Landscape Analysis: CDGA Leads to a Flatter Local Minimum

Recently, there has been a growing interest in finding the relation between loss landscape structure and generalization ability of deep neural networks (Keskar et al., 2016; Foret et al., 2020). More precisely, according to the theorem stated by Foret et al. (2020) generalization error of deep neural networks is upper bounded by the sharpness of their loss landscapes:

$$\max_{\|\epsilon\|_2 \leq \rho} \mathcal{L}(\theta + \epsilon) - \mathcal{L}(\theta), \tag{9}$$

where $\rho$ is the perturbation size. In other words, neural networks with flatter local minima have less generalization error. Inspired by this finding, we raise the following question. Given better OOD generalization of CDGA and CDGA*, do these methods lead to flatter local minima compared with ERM? To answer this question we calculate the sharpness using Eq. 9 and monitor it through the training of the model on the PACS dataset. The result of this experiment is reported in Figure 9 (right). Interestingly, CDGA and CDGA* lead to less sharp local minima, which explains their superior generalization ability.

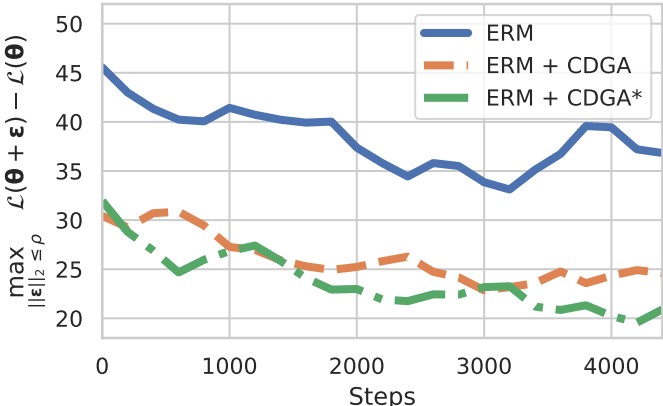

Figure 9: Sharpness of loss landscape through training for ERM, CDGA, and CDGA* on PACS dataset.

## A.2    CDGA Improves Adversarial Robustness

The robustness of deep learning models to adversarial attacks is a notion of their generalization ability (Shorten & Khoshgoftaar, 2019). Here, we study the robustness of models trained with CDGA and CDGA* against adversarial shifts to the data. To this end, we evaluate their performance on the target domain after shifting the target domain using the fast gradient sign method (FGSM) (Goodfellow et al., 2015), and the more sophisticated projected gradient descent (PGD) adversarial attacks (Madry et al., 2018). We design an experiment using the PACS dataset where the training domains are P, A, and C and the target domain is S. After training the same model using different data augmentation techniques including MixUp and the classic data augmentation, we perturb the target domain using FSGM and PGD and record the OOD accuracy for

different strength of attacks. Results are presented in Figure 10, where $\rho$ controls the FGSM perturbation size, and $K$ controls the number of iterations in PGD. Clearly, models trained with CDGA and CDGA* are more robust against adversarial shifts on the target domain compared with the classic data augmentation and Mixup.

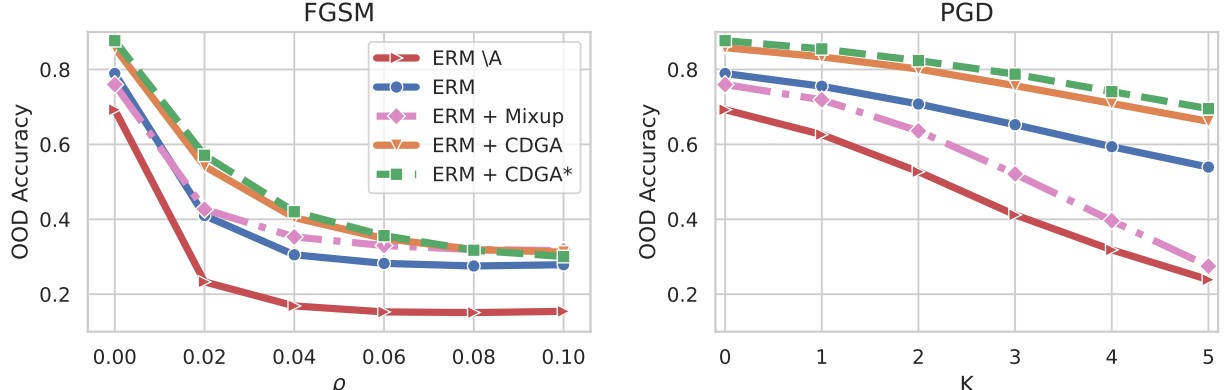

Figure 10: OOD accuracy of models trained using CDGA, CDGA*, classic augmentation, and MixUp after FGSM adversarial shift with varying perturbation size $\rho$ (left Figure) and PGD adversarial shift with varying number of iterations $K$ (right Figure). Models trained with CDGA and CDGA* are more robust against adversarial shifts. ERM \A means without data augmentation.

## B  Visualization of Different Generative Augmentation Algorithm Results

In Figure 11, we provide three examples that are augmented using different variations of SDGA and CDGA.

## C  Test Domain Near-duplication Analysis Full Results

To quantify how much the generated images are similar to the original images for each domain, in section 6, Figures 6 and 7 we presented the summarized results for near-duplicate image detection. More precisely, near-duplicate image detection was applied to images generated using CDGA to quantify how much the generated images are similar to the original images for each domain. Here, we present the extended version of these results in Figures 12 and 13 respectively. In Figure 12, for each original domain, we report the percentage of near-duplicates over the size of the original domain. Clearly, generating synthetic images that exist in the manifold between training domains enables us to have examples near-duplicate to the target domain. Figure 13 shows multiple examples where the synthetically generated images are near-duplicates to real data. These examples show how CDGA can reduce the domain shift between training domains and the target domain.

## D  Transferability Full Results

In section 6, we monitored classifier heads' Hessian distances as a measure of domain shift and transferability. We calculated classifier heads' Hessian distances between all possible domain pairs through the training steps for ERM, CDGA, and CDGA* where we set domains P, A, and C as train (source) and domain S as the target. Figure 5, shows the difference between the classifier head's Hessians given data from domains A and S during the steps. For the sake of completeness, here we extend this result and present classifier heads' Hessian distances between all domain pairs namely (A,S), (P,S), and (C,S) in Figure 14. Clearly, in all cases, The distance between classifier heads for CDGA and CDGA* is smaller compared with the case where only real data is used (ERM).

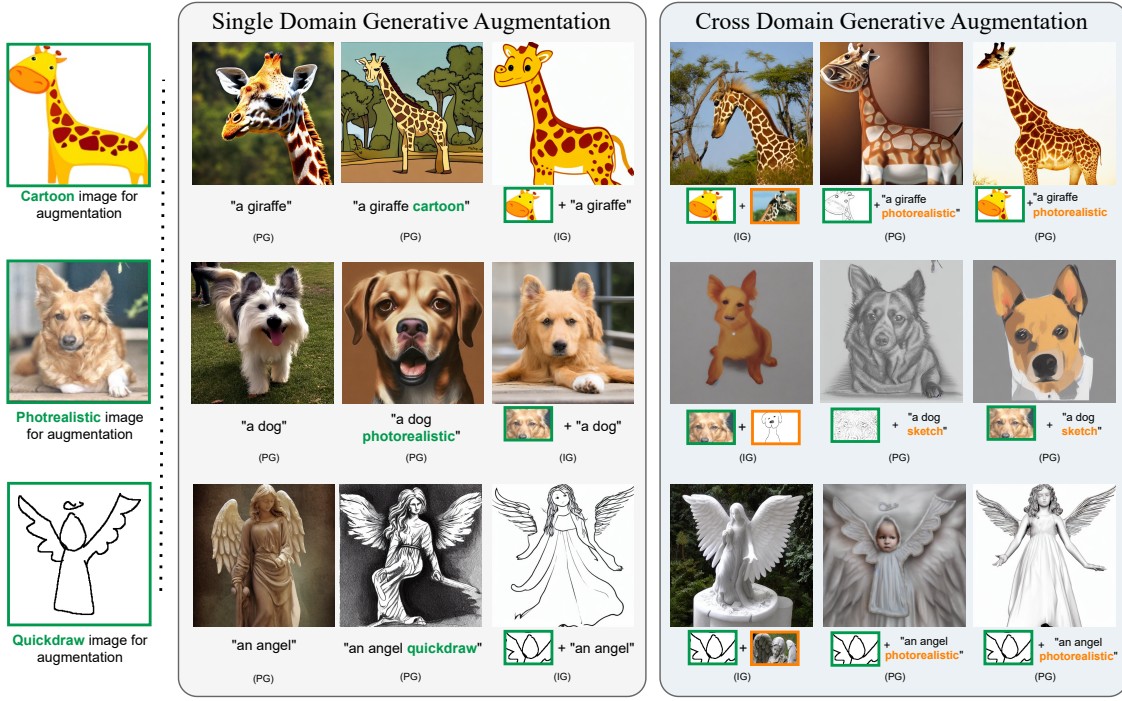

Figure 11: We compare different CDGA and SDGA techniques. The left column shows the original images intended for generative augmentation. The middle and right columns show different variations of SDGA and CDGA respectively with their prompts.

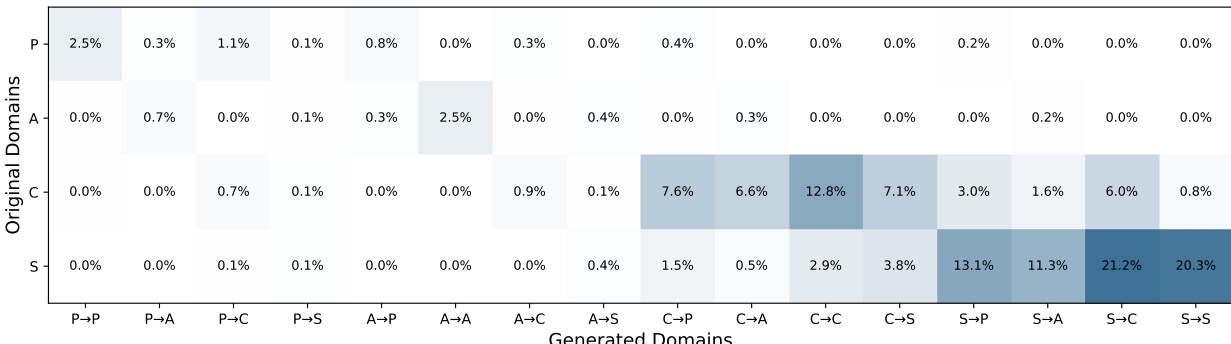

Figure 12: Heat map of the number of near-duplicates of each target domain that are in each original and generated dataset. This table shows that using test-domain description results in more near-duplicate images.

## E  $t$-SNE plots

In Figure 4, we presented a 2D projection of the original PACS dataset from all domains along with CDGA-based data obtained from Domain A only for the "Dog" class. This figure showed how the cross-domain synthetic images interpolate different domains as we desired. Here in Figure 15, we present the results of this experiment for all other classes in the PACS dataset. As can be seen, for most classes the synthetic examples consistently reduce the domain shift which results in better OOD performance of ERM.

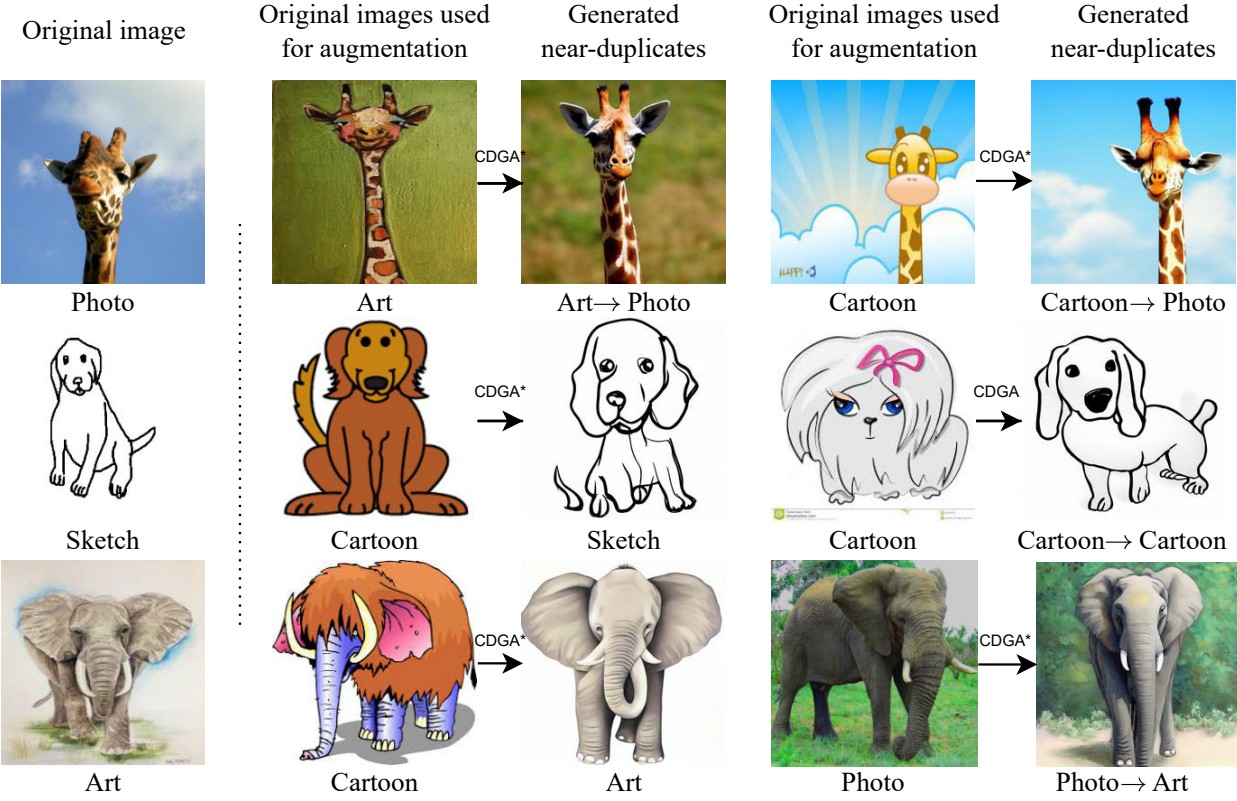

Figure 13: Illustration of near-duplicates of three images from the test domain (left-most column) that are generated using cross-domain generative augmentation (denoted by the arrow) from the original images and are in the training domain.

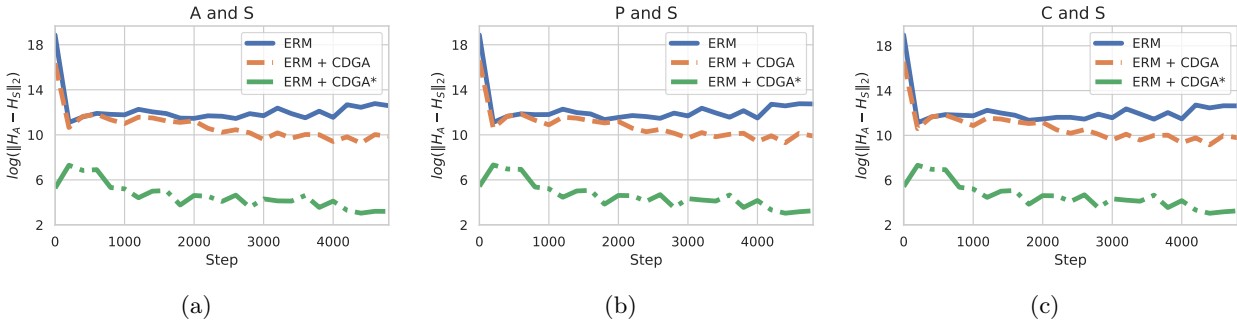

Figure 14: Classifier head Hessian difference during training for PACS and Cross Domain generative Augmented PACS datasets i.e., CDGA-PACS and CDGA*-PACS where Domains A P, and C are training domains and domain S is the target domain.

## F  DomainBed benchmark full results

To save space in the main paper, for the DomainBed results in Tables 1- 4 we only reported the five top-performing methods for each model selection technique. Here in Tables 8, 9, 10, and 11 we present the results for all algorithms that have been tested on the DomainBed benchmark (Rame et al., 2022; Gulrajani & Lopez-Paz, 2020). Given that all the results presented for the DomainBed so far are averaged performances for

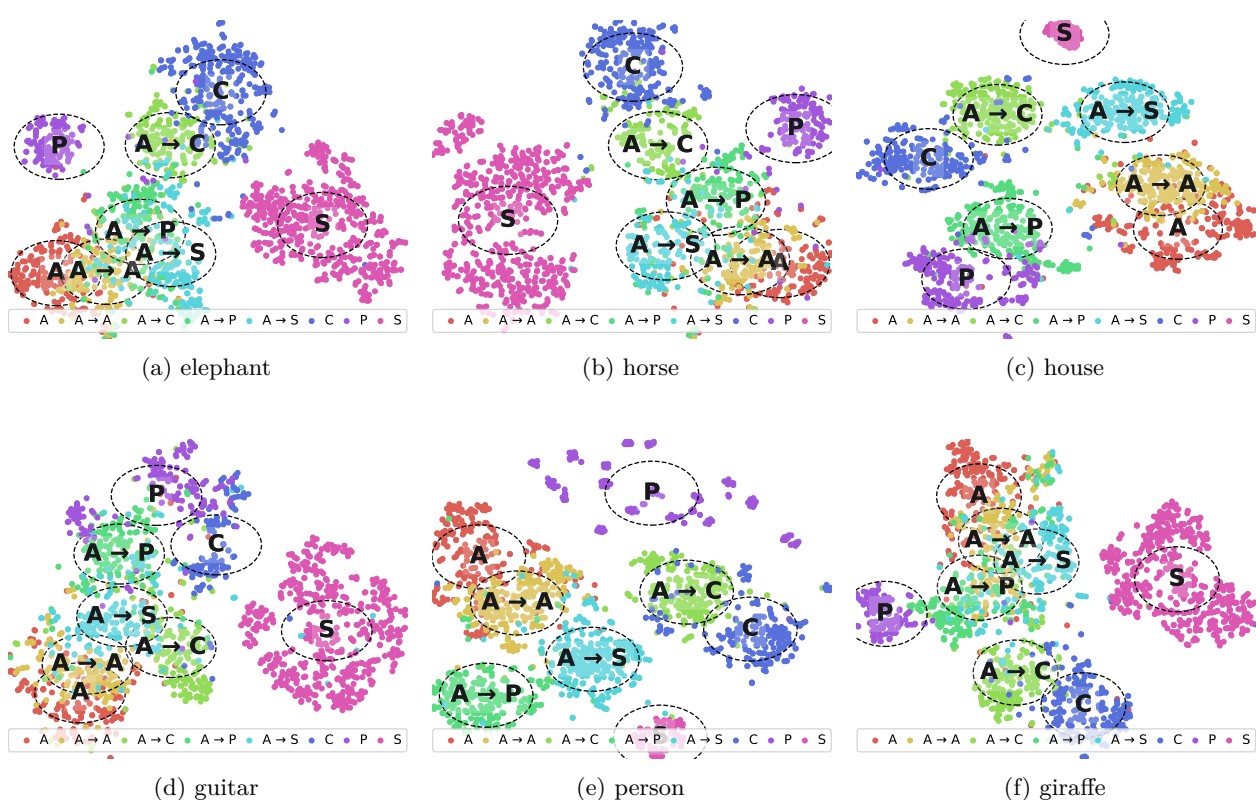

(a) elephant        (b) horse        (c) house

(d) guitar        (e) person        (f) giraffe

Figure 15: The $t$-SNE plot of features extracted from the original PACS dataset and generated images using CDGA by the LDM from A domain for all classes. This figure shows that CDGA can fill the gap between domains.

the leave-one-domain-out experiments. The detailed per-domain results for PACS, OfficeHome, DomainNet, and VLCS are presented in Tables 12, 13, 14, 15, 16, 17, 18, 19, 20, 21, 22, and 23.

Table 8: DomainBed benchmark for **training-domain validation set** model selection method. We format **first**, second and worse than ERM results.

| Algorithm | PACS | OfficeHome | DomainNet | Avg |
|---|---|---|---|---|
| ERM | 85.5 ± 0.2 | 66.5 ± 0.3 | 40.9 ± 1.8 | 64.3 |
| IRM | 83.5 ± 0.8 | 64.3 ± 2.2 | 33.9 ± 2.8 | 60.6 |
| GroupDRO | 84.4 ± 0.8 | 66.0 ± 0.7 | 33.3 ± 0.2 | 61.2 |
| Mixup | 84.6 ± 0.6 | 68.1 ± 0.3 | 39.2 ± 0.1 | 64.0 |
| MLDG | 84.9 ± 1.0 | 66.8 ± 0.6 | 41.2 ± 0.1 | 64.3 |
| CORAL | 86.2 ± 0.3 | 68.7 ± 0.3 | 41.5 ± 0.1 | 65.5 |
| MMD | 84.6 ± 0.5 | 66.3 ± 0.1 | 23.4 ± 9.5 | 58.1 |
| DANN | 83.6 ± 0.4 | 65.9 ± 0.6 | 38.3 ± 0.1 | 62.6 |
| CDANN | 82.6 ± 0.9 | 65.8 ± 1.3 | 38.3 ± 0.3 | 62.2 |
| MTL | 84.6 ± 0.5 | 66.4 ± 0.5 | 40.6 ± 0.1 | 63.9 |
| SagNet | 86.3 ± 0.2 | 68.1 ± 0.1 | 40.3 ± 0.1 | 64.9 |
| ARM | 85.1 ± 0.4 | 64.8 ± 0.3 | 35.5 ± 0.2 | 61.8 |
| V-REx | 84.9 ± 0.6 | 66.4 ± 0.6 | 33.6 ± 2.9 | 61.6 |
| RSC | 85.2 ± 0.9 | 65.5 ± 0.9 | 38.9 ± 0.5 | 63.2 |
| AND-mask | 84.4 ± 0.9 | 65.6 ± 0.4 | 37.2 ± 0.6 | 62.4 |
| SAND-mask | 84.6 ± 0.9 | 65.8 ± 0.4 | 32.1 ± 0.6 | 60.8 |
| Fish | 85.5 ± 0.3 | 68.6 ± 0.4 | 42.7 ± 0.2 | 65.6 |
| Fishr | 85.5 ± 0.4 | 67.8 ± 0.1 | 41.7 ± 0.0 | 65.0 |
| HGP | 84.7 ± 0.0 | 68.2 ± 0.0 | 41.1 ± 0.0 | 64.7 |
| Hutchinson | 83.9 ± 0.0 | 68.2 ± 0.0 | 41.6 ± 0.0 | 64.6 |
| ERM + CDGA-PG | 88.5 ± 0.5 | 68.2 ± 0.6 | 43.1 ±0.0 | 66.6 |
| ERM + CDGA-PG* | **89.5** ± 0.3 | **70.8** ± 0.6 | **44.8** ±0.0 | **68.4** |

Table 9: DomainBed benchmark for **leave-one-domain-out cross-validation** model selection. We format **first**, second and worse than ERM results.

| Algorithm | PACS | OfficeHome | DomainNet | Avg |
|---|---|---|---|---|
| ERM | 83.0 ± 0.7 | 65.7 ± 0.5 | 40.6 ± 0.2 | 63.1 |
| IRM | 81.5 ± 0.8 | 64.3 ± 1.5 | 33.5 ± 0.3 | 59.8 |
| GroupDRO | 83.5 ± 0.2 | 65.2 ± 0.2 | 33.0 ± 0.3 | 60.6 |
| Mixup | 83.2 ± 0.4 | 67.0 ± 0.2 | 38.5 ± 0.3 | 62.9 |
| MLDG | 82.9 ± 1.7 | 66.1 ± 0.5 | 41.0 ± 0.2 | 63.3 |
| CORAL | 82.6 ± 0.5 | 68.5 ± 0.2 | 41.1 ± 0.1 | 64.1 |
| MMD | 83.2 ± 0.2 | 60.2 ± 5.2 | 23.4 ± 9.5 | 55.6 |
| DANN | 81.0 ± 1.1 | 64.9 ± 1.2 | 38.2 ± 0.2 | 61.4 |
| CDANN | 78.8 ± 2.2 | 64.3 ± 1.7 | 38.0 ± 0.1 | 60.4 |
| MTL | 83.7 ± 0.4 | 65.7 ± 0.5 | 40.6 ± 0.1 | 63.3 |
| SagNet | 82.3 ± 0.1 | 67.6 ± 0.3 | 40.2 ± 0.2 | 63.4 |
| ARM | 81.7 ± 0.2 | 64.4 ± 0.2 | 35.2 ± 0.1 | 60.4 |
| V-REx | 81.3 ± 0.9 | 64.9 ± 1.3 | 33.4 ± 3.1 | 59.9 |
| RSC | 82.6 ± 0.7 | 65.8 ± 0.7 | 38.9 ±0.5 | 62.4 |
| HGP | 82.2 ± 0.0 | 67.5 ± 0.0 | 41.1 ± 0.0 | 63.6 |
| Hutchinson | 84.8 ± 0.0 | 68.5 ± 0.0 | 41.4 ± 0.0 | 64.9 |
| ERM + CDGA-PG | 86.8 ±0.4 | 68.7 ± 0.4 | 43.1 ± 0.0 | 66.2 |
| ERM + CDGA*-PG | **88.4** ±0.5 | **70.2** ± 0.4 | **44.8** ± 0.0 | **67.8** |

Table 10: DomainBed benchmark for **test-domain validation set (oracle)** model selection method. We format **first**, second and worse than ERM results.

| Algorithm | PACS | OfficeHome | DomainNet | Avg |
|---|---|---|---|---|
| ERM | $86.7 \pm 0.3$ | $66.4 \pm 0.5$ | $41.3 \pm 0.1$ | 64.8 |
| IRM | $84.5 \pm 1.1$ | $63.0 \pm 2.7$ | $28.0 \pm 5.1$ | 58.5 |
| GroupDRO | $87.1 \pm 0.1$ | $66.2 \pm 0.6$ | $33.4 \pm 0.3$ | 62.2 |
| Mixup | $86.8 \pm 0.3$ | $68.0 \pm 0.2$ | $39.6 \pm 0.1$ | 64.8 |
| MLDG | $86.8 \pm 0.4$ | $66.6 \pm 0.3$ | $41.6 \pm 0.1$ | 65.0 |
| CORAL | $87.1 \pm 0.5$ | $68.4 \pm 0.2$ | $41.8 \pm 0.1$ | 65.8 |
| MMD | $87.2 \pm 0.1$ | $66.2 \pm 0.3$ | $23.5 \pm 9.4$ | 59.0 |
| DANN | $85.2 \pm 0.2$ | $65.3 \pm 0.8$ | $38.3 \pm 0.1$ | 62.9 |
| CDANN | $85.8 \pm 0.8$ | $65.3 \pm 0.5$ | $38.5 \pm 0.2$ | 63.2 |
| MTL | $86.7 \pm 0.2$ | $66.5 \pm 0.4$ | $40.8 \pm 0.1$ | 64.7 |
| SagNet | $86.4 \pm 0.4$ | $67.5 \pm 0.2$ | $40.8 \pm 0.2$ | 64.9 |
| ARM | $85.8 \pm 0.2$ | $64.8 \pm 0.4$ | $36.0 \pm 0.2$ | 62.2 |
| V-REx | $87.2 \pm 0.6$ | $65.7 \pm 0.3$ | $30.1 \pm 3.7$ | 61.0 |
| RSC | $86.2 \pm 0.5$ | $66.5 \pm 0.6$ | $38.9 \pm 0.6$ | 63.9 |
| AND-mask | $86.4 \pm 0.4$ | $66.1 \pm 0.2$ | $37.9 \pm 0.6$ | 63.5 |
| SAND-mask | $85.9 \pm 0.4$ | $65.9 \pm 0.5$ | $32.2 \pm 0.6$ | 61.3 |
| Fish | $85.5 \pm 0.3$ | $68.6 \pm 0.4$ | $42.7 \pm 0.2$ | 65.6 |
| Fishr | $85.8 \pm 0.6$ | $66.0 \pm 2.9$ | $43.4 \pm 0.3$ | 65.1 |
| Hutchinson | $86.3 \pm 0.0$ | $68.4 \pm 0.0$ | $41.9 \pm 0.0$ | 65.5 |
| HGP | $86.5 \pm 0.0$ | $67.4 \pm 0.0$ | $41.2 \pm 0.0$ | 65.0 |
| ERM + CDGA-PG | 89.6 $\pm 0.3$ | 68.8 $\pm 0.3$ | 43.1 $\pm 0.0$ | 67.2 |
| ERM + CDGA*-PG | **90.4** $\pm 0.3$ | **70.2** $\pm 0.2$ | **44.8** $\pm 0.0$ | **68.5** |

Table 11: DomainBed benchmark on **VLCS** dataset across different model selection methods. We format **first**, second and worse than ERM results.

| Method | Training domain | Leave-one-domain-out | Oracle |
|---|---|---|---|
| ERM | $77.5 \pm 0.4$ | $77.2 \pm 0.4$ | $77.6 \pm 0.3$ |
| IRM | $78.5 \pm 0.5$ | $76.3 \pm 0.6$ | $76.9 \pm 0.6$ |
| GroupDRO | $76.7 \pm 0.6$ | $77.9 \pm 0.5$ | $77.4 \pm 0.5$ |
| Mixup | $77.4 \pm 0.6$ | $77.7 \pm 0.6$ | $78.1 \pm 0.3$ |
| MLDG | $77.2 \pm 0.4$ | $77.2 \pm 0.9$ | $77.5 \pm 0.1$ |
| CORAL | 78.8 $\pm 0.6$ | 78.7 $\pm 0.4$ | $77.7 \pm 0.2$ |
| MMD | $77.5 \pm 0.9$ | $77.3 \pm 0.5$ | $77.9 \pm 0.1$ |
| DANN | $78.6 \pm 0.4$ | $76.9 \pm 0.4$ | $79.7 \pm 0.5$ |
| CDANN | $77.5 \pm 0.1$ | $77.5 \pm 0.2$ | 79.9 $\pm 0.2$ |
| MTL | $77.2 \pm 0.4$ | $76.6 \pm 0.5$ | $77.7 \pm 0.5$ |
| SagNet | $77.8 \pm 0.5$ | $77.5 \pm 0.3$ | $77.6 \pm 0.1$ |
| Fishr | $77.8 \pm 0.1$ | $78.2 \pm 0.0$ | 78.2 $\pm 0.2$ |
| HGP | $77.6 \pm 0.0$ | $76.7 \pm 0.0$ | $77.3 \pm 0.0$ |
| Hutchinson | $76.8 \pm 0.0$ | **79.3** $\pm 0.0$ | $77.9 \pm 0.0$ |
| ERM + CDGA-IG | **78.9** $\pm 0.3$ | $77.9 \pm 0.5$ | **79.5** $\pm 0.1$ |

Table 12: DomainBed benchmark, **PACS full results for training-domain validation set** model selection method.

| Algorithm | A | C | P | S | Avg |
|---|---|---|---|---|---|
| ERM | 84.7 ± 0.4 | 80.8 ± 0.6 | 97.2 ± 0.3 | 79.3 ± 1.0 | 85.5 |
| IRM | 84.8 ± 1.3 | 76.4 ± 1.1 | 96.7 ± 0.6 | 76.1 ± 1.0 | 83.5 |
| GroupDRO | 83.5 ± 0.9 | 79.1 ± 0.6 | 96.7 ± 0.3 | 78.3 ± 2.0 | 84.4 |
| Mixup | 86.1 ± 0.5 | 78.9 ± 0.8 | 97.6 ± 0.1 | 75.8 ± 1.8 | 84.6 |
| MLDG | 85.5 ± 1.4 | 80.1 ± 1.7 | 97.4 ± 0.3 | 76.6 ± 1.1 | 84.9 |
| CORAL | 88.3 ± 0.2 | 80.0 ± 0.5 | 97.5 ± 0.3 | 78.8 ± 1.3 | 86.2 |
| MMD | 86.1 ± 1.4 | 79.4 ± 0.9 | 96.6 ± 0.2 | 76.5 ± 0.5 | 84.6 |
| DANN | 86.4 ± 0.8 | 77.4 ± 0.8 | 97.3 ± 0.4 | 73.5 ± 2.3 | 83.6 |
| CDANN | 84.6 ± 1.8 | 75.5 ± 0.9 | 96.8 ± 0.3 | 73.5 ± 0.6 | 82.6 |
| MTL | 87.5 ± 0.8 | 77.1 ± 0.5 | 96.4 ± 0.8 | 77.3 ± 1.8 | 84.6 |
| SagNet | 87.4 ± 1.0 | 80.7 ± 0.6 | 97.1 ± 0.1 | 80.0 ± 0.4 | 86.3 |
| ARM | 86.8 ± 0.6 | 76.8 ± 0.5 | 97.4 ± 0.3 | 79.3 ± 1.2 | 85.1 |
| V-REx | 86.0 ± 1.6 | 79.1 ± 0.6 | 96.9 ± 0.5 | 77.7 ± 1.7 | 84.9 |
| RSC | 85.4 ± 0.8 | 79.7 ± 1.8 | **97.6** ± 0.3 | 78.2 ± 1.2 | 85.2 |
| AND-mask | 85.3 ± 1.4 | 79.2 ± 2.0 | 96.9 ± 0.4 | 76.2 ± 1.4 | 84.4 |
| SAND-mask | 85.8 ± 1.7 | 79.2 ± 0.8 | 96.3 ± 0.2 | 76.9 ± 2.0 | 84.6 |
| Fish | - | - | - | - | 85.5 |
| Fishr | 88.4 ± 0.2 | 78.7 ± 0.7 | 97.0 ± 0.1 | 77.8 ± 2.0 | 85.5 |
| CDGA-PG | 89.1 ± 1.0 | 82.5 ± 0.5 | 97.4 ± 0.2 | **84.8** ± 0.9 | 88.5 |
| CDGA*-PG | **89.7** ± 1.1 | **86.6** ± 0.3 | 97.4 ± 0.1 | 84.3 ± 1.6 | **89.5** |

Table 13: DomainBed benchmark, **PACS full results for leave-one-domain-out cross-validation** model selection method.

| Algorithm | A | C | P | S | Avg |
|---|---|---|---|---|---|
| ERM | 83.2 ± 1.3 | 76.8 ± 1.7 | **97.2** ± 0.3 | 74.8 ± 1.3 | 83.0 |
| IRM | 81.7 ± 2.4 | 77.0 ± 1.3 | 96.3 ± 0.2 | 71.1 ± 2.2 | 81.5 |
| GroupDRO | 84.4 ± 0.7 | 77.3 ± 0.8 | 96.8 ± 0.8 | 75.6 ± 1.4 | 83.5 |
| Mixup | 85.2 ± 1.9 | 77.0 ± 1.7 | 96.8 ± 0.8 | 73.9 ± 1.6 | 83.2 |
| MLDG | 81.4 ± 3.6 | 77.9 ± 2.3 | 96.2 ± 0.3 | 76.1 ± 2.1 | 82.9 |
| CORAL | 80.5 ± 2.8 | 74.5 ± 0.4 | 96.8 ± 0.3 | 78.6 ± 1.4 | 82.6 |
| MMD | 84.9 ± 1.7 | 75.1 ± 2.0 | 96.1 ± 0.9 | 76.5 ± 1.5 | 83.2 |
| DANN | 84.3 ± 2.8 | 72.4 ± 2.8 | 96.5 ± 0.8 | 70.8 ± 1.3 | 81.0 |
| CDANN | 78.3 ± 2.8 | 73.8 ± 1.6 | 96.4 ± 0.5 | 66.8 ± 5.5 | 78.8 |
| MTL | 85.6 ± 1.5 | 78.9 ± 0.6 | 97.1 ± 0.3 | 73.1 ± 2.7 | 83.7 |
| SagNet | 81.1 ± 1.9 | 75.4 ± 1.3 | 95.7 ± 0.9 | 77.2 ± 0.6 | 82.3 |
| ARM | 85.9 ± 0.3 | 73.3 ± 1.9 | 95.6 ± 0.4 | 72.1 ± 2.4 | 81.7 |
| VREx | 81.6 ± 4.0 | 74.1 ± 0.3 | 96.9 ± 0.4 | 72.8 ± 2.1 | 81.3 |
| RSC | 83.7 ± 1.7 | 82.9 ± 1.1 | 95.6 ± 0.7 | 68.1 ± 1.5 | 82.6 |
| CDGA-PG | 87.3 ± 1.5 | 80.9 ± 1.6 | 96.6 ± 0.7 | **82.5** ± 0.9 | 86.8 |
| CDGA*-PG | **88.1** ± 1.1 | **86.6** ± 1.0 | **97.2** ± 0.4 | 81.9 ± 1.0 | **88.4** |

Table 14: DomainBed benchmark, **PACS full results for test-domain validation set (oracle)** model selection method.

| Algorithm | A | C | P | S | Avg |
|---|---|---|---|---|---|
| ERM | $86.5 \pm 1.0$ | $81.3 \pm 0.6$ | $96.2 \pm 0.3$ | $82.7 \pm 1.1$ | 86.7 |
| IRM | $84.2 \pm 0.9$ | $79.7 \pm 1.5$ | $95.9 \pm 0.4$ | $78.3 \pm 2.1$ | 84.5 |
| GroupDRO | $87.5 \pm 0.5$ | $82.9 \pm 0.6$ | $97.1 \pm 0.3$ | $81.1 \pm 1.2$ | 87.1 |
| Mixup | $87.5 \pm 0.4$ | $81.6 \pm 0.7$ | $\underline{97.4} \pm 0.2$ | $80.8 \pm 0.9$ | 86.8 |
| MLDG | $87.0 \pm 1.2$ | $82.5 \pm 0.9$ | $96.7 \pm 0.3$ | $81.2 \pm 0.6$ | 86.8 |
| CORAL | $86.6 \pm 0.8$ | $81.8 \pm 0.9$ | $97.1 \pm 0.5$ | $82.7 \pm 0.6$ | 87.1 |
| MMD | $88.1 \pm 0.8$ | $82.6 \pm 0.7$ | $97.1 \pm 0.5$ | $81.2 \pm 1.2$ | 87.2 |
| DANN | $87.0 \pm 0.4$ | $80.3 \pm 0.6$ | $96.8 \pm 0.3$ | $76.9 \pm 1.1$ | 85.2 |
| CDANN | $87.7 \pm 0.6$ | $80.7 \pm 1.2$ | $97.3 \pm 0.4$ | $77.6 \pm 1.5$ | 85.8 |
| MTL | $87.0 \pm 0.2$ | $82.7 \pm 0.8$ | $96.5 \pm 0.7$ | $80.5 \pm 0.8$ | 86.7 |
| SagNet | $87.4 \pm 0.5$ | $81.2 \pm 1.2$ | $96.3 \pm 0.8$ | $80.7 \pm 1.1$ | 86.4 |
| ARM | $85.0 \pm 1.2$ | $81.4 \pm 0.2$ | $95.9 \pm 0.3$ | $80.9 \pm 0.5$ | 85.8 |
| V-REx | $87.8 \pm 1.2$ | $81.8 \pm 0.7$ | $\underline{97.4} \pm 0.2$ | $82.1 \pm 0.7$ | 87.2 |
| RSC | $86.0 \pm 0.7$ | $81.8 \pm 0.9$ | $96.8 \pm 0.7$ | $80.4 \pm 0.5$ | 86.2 |
| AND-mask | $86.4 \pm 1.1$ | $80.8 \pm 0.9$ | $97.1 \pm 0.2$ | $81.3 \pm 1.1$ | 86.4 |
| SAND-mask | $86.1 \pm 0.6$ | $80.3 \pm 1.0$ | $97.1 \pm 0.3$ | $80.0 \pm 1.3$ | 85.9 |
| Fish | - | - | - | - | 85.8 |
| Fishr | $87.9 \pm 0.6$ | $80.8 \pm 0.5$ | $\mathbf{97.9} \pm 0.4$ | $81.1 \pm 0.8$ | 86.9 |
| CDGA-PG | $\underline{89.6} \pm 0.8$ | $\underline{85.3} \pm 0.7$ | $97.3 \pm 0.3$ | $\mathbf{86.2} \pm 0.5$ | $\underline{89.6}$ |
| CDGA*-PG | $\mathbf{90.3} \pm 0.8$ | $\mathbf{89.0} \pm 0.2$ | $96.8 \pm 0.1$ | $\underline{85.7} \pm 1.0$ | $\mathbf{90.4}$ |

Table 15: DomainBed benchmark, **OfficeHome full results for training-domain validation set** model selection method.

| Algorithm | A | C | P | R | Avg |
|---|---|---|---|---|---|
| ERM | $61.3 \pm 0.7$ | $52.4 \pm 0.3$ | $75.8 \pm 0.1$ | $76.6 \pm 0.3$ | 66.5 |
| IRM | $58.9 \pm 2.3$ | $52.2 \pm 1.6$ | $72.1 \pm 2.9$ | $74.0 \pm 2.5$ | 64.3 |
| GroupDRO | $60.4 \pm 0.7$ | $52.7 \pm 1.0$ | $75.0 \pm 0.7$ | $76.0 \pm 0.7$ | 66.0 |
| Mixup | $62.4 \pm 0.8$ | $\underline{54.8} \pm 0.6$ | $\mathbf{76.9} \pm 0.3$ | $78.3 \pm 0.2$ | 68.1 |
| MLDG | $61.5 \pm 0.9$ | $53.2 \pm 0.6$ | $75.0 \pm 1.2$ | $77.5 \pm 0.4$ | 66.8 |
| CORAL | $\mathbf{65.3} \pm 0.4$ | $54.4 \pm 0.5$ | $\underline{76.5} \pm 0.1$ | $78.4 \pm 0.5$ | $\underline{68.7}$ |
| MMD | $60.4 \pm 0.2$ | $53.3 \pm 0.3$ | $74.3 \pm 0.1$ | $77.4 \pm 0.6$ | 66.3 |
| DANN | $59.9 \pm 1.3$ | $53.0 \pm 0.3$ | $73.6 \pm 0.7$ | $76.9 \pm 0.5$ | 65.9 |
| CDANN | $61.5 \pm 1.4$ | $50.4 \pm 2.4$ | $74.4 \pm 0.9$ | $76.6 \pm 0.8$ | 65.8 |
| MTL | $61.5 \pm 0.7$ | $52.4 \pm 0.6$ | $74.9 \pm 0.4$ | $76.8 \pm 0.4$ | 66.4 |
| SagNet | $\underline{63.4} \pm 0.2$ | $\underline{54.8} \pm 0.4$ | $75.8 \pm 0.4$ | $\underline{78.3} \pm 0.3$ | 68.1 |
| ARM | $58.9 \pm 0.8$ | $51.0 \pm 0.5$ | $74.1 \pm 0.1$ | $75.2 \pm 0.3$ | 64.8 |
| V-REx | $60.7 \pm 0.9$ | $53.0 \pm 0.9$ | $75.3 \pm 0.1$ | $76.6 \pm 0.5$ | 66.4 |
| RSC | $60.7 \pm 1.4$ | $51.4 \pm 0.3$ | $74.8 \pm 1.1$ | $75.1 \pm 1.3$ | 65.5 |
| ANDMask | $59.5 \pm 1.2$ | $51.7 \pm 0.2$ | $73.9 \pm 0.4$ | $77.1 \pm 0.2$ | 65.6 |
| SAND-mask | $60.3 \pm 0.5$ | $53.3 \pm 0.7$ | $73.5 \pm 0.7$ | $76.2 \pm 0.3$ | 65.8 |
| Fish | - | - | - | - | 68.6 |
| Fishr | $62.4 \pm 0.5$ | $54.4 \pm 0.4$ | $76.2 \pm 0.5$ | $\underline{78.3} \pm 0.1$ | 67.8 |
| CDGA-PG | $60.1 \pm 1.4$ | $54.2 \pm 0.5$ | $78.2 \pm 0.6$ | $\underline{80.4} \pm 0.1$ | 68.2 |
| CDGA*-PG | $\mathbf{63.1} \pm 1.5$ | $\mathbf{60.2} \pm 0.1$ | $79.4 \pm 0.7$ | $\mathbf{80.5} \pm 0.2$ | $\mathbf{70.8}$ |

Table 16: DomainBed benchmark, **OfficeHome full results for leave-one-domain-out cross-validation** model selection method.

| Algorithm | A | C | P | R | Avg |
|---|---|---|---|---|---|
| ERM | 61.1 ± 0.9 | 50.7 ± 0.6 | 74.6 ± 0.3 | 76.4 ± 0.6 | 65.7 |
| IRM | 58.2 ± 1.2 | 51.6 ± 1.2 | 73.3 ± 2.2 | 74.1 ± 1.7 | 64.3 |
| GroupDRO | 59.9 ± 0.4 | 51.0 ± 0.4 | 73.7 ± 0.3 | 76.0 ± 0.2 | 65.2 |
| Mixup | 61.4 ± 0.5 | 53.0 ± 0.3 | 75.8 ± 0.2 | 77.7 ± 0.3 | 67.0 |
| MLDG | 60.5 ± 1.4 | 51.9 ± 0.2 | 74.4 ± 0.6 | 77.6 ± 0.4 | 66.1 |
| CORAL | 64.5 ± 0.8 | 54.8 ± 0.2 | 76.6 ± 0.3 | 78.1 ± 0.2 | 68.5 |
| MMD | 60.8 ± 0.7 | 53.7 ± 0.5 | 50.2 ± 19.9 | 76.0 ± 0.7 | 60.2 |
| DANN | 60.2 ± 1.3 | 52.2 ± 0.9 | 71.3 ± 2.0 | 76.0 ± 0.6 | 64.9 |
| CDANN | 58.7 ± 2.9 | 49.0 ± 2.1 | 73.6 ± 1.0 | 76.0 ± 1.1 | 64.3 |
| MTL | 59.1 ± 0.3 | 52.1 ± 1.2 | 74.7 ± 0.4 | 77.0 ± 0.6 | 65.7 |
| SagNet | **63.0** ± 0.8 | 54.0 ± 0.3 | 76.6 ± 0.3 | 76.8 ± 0.4 | 67.6 |
| ARM | 58.7 ± 0.8 | 49.8 ± 1.1 | 73.1 ± 0.5 | 75.9 ± 0.1 | 64.4 |
| VREx | 57.6 ± 3.4 | 51.3 ± 1.3 | 74.9 ± 0.2 | 75.8 ± 0.7 | 64.9 |
| RSC | 61.6 ± 1.0 | 51.1 ± 0.8 | 74.8 ± 1.1 | 75.7 ± 0.9 | 65.8 |
| CDGA-PG | 60.5 ± 1.2 | 56.5 ± 0.3 | 77.1 ± 0.4 | **80.6** ± 0.2 | 68.7 |
| CDGA*-PG | 62.9 ± 0.4 | **59.9** ± 0.5 | **78.1** ± 0.9 | 79.9 ± 0.4 | **70.2** |

Table 17: DomainBed benchmark, **OfficeHome full results for test-domain validation set (oracle)** model selection method.

| Algorithm | A | C | P | R | Avg |
|---|---|---|---|---|---|
| ERM | 61.7 ± 0.7 | 53.4 ± 0.3 | 74.1 ± 0.4 | 76.2 ± 0.6 | 66.4 |
| IRM | 56.4 ± 3.2 | 51.2 ± 2.3 | 71.7 ± 2.7 | 72.7 ± 2.7 | 63.0 |
| GroupDRO | 60.5 ± 1.6 | 53.1 ± 0.3 | 75.5 ± 0.3 | 75.9 ± 0.7 | 66.2 |
| Mixup | 63.5 ± 0.2 | 54.6 ± 0.4 | 76.0 ± 0.3 | 78.0 ± 0.7 | 68.0 |
| MLDG | 60.5 ± 0.7 | 54.2 ± 0.5 | 75.0 ± 0.2 | 76.7 ± 0.5 | 66.6 |
| CORAL | **64.8** ± 0.8 | 54.1 ± 0.9 | 76.5 ± 0.4 | 78.2 ± 0.4 | 68.4 |
| MMD | 60.4 ± 1.0 | 53.4 ± 0.5 | 74.9 ± 0.1 | 76.1 ± 0.7 | 66.2 |
| DANN | 60.6 ± 1.4 | 51.8 ± 0.7 | 73.4 ± 0.5 | 75.5 ± 0.9 | 65.3 |
| CDANN | 57.9 ± 0.2 | 52.1 ± 1.2 | 74.9 ± 0.7 | 76.2 ± 0.2 | 65.3 |
| MTL | 60.7 ± 0.8 | 53.5 ± 1.3 | 75.2 ± 0.6 | 76.6 ± 0.6 | 66.5 |
| SagNet | 62.7 ± 0.5 | 53.6 ± 0.5 | 76.0 ± 0.3 | 77.8 ± 0.1 | 67.5 |
| ARM | 58.8 ± 0.5 | 51.8 ± 0.7 | 74.0 ± 0.1 | 74.4 ± 0.2 | 64.8 |
| V-REx | 59.6 ± 1.0 | 53.3 ± 0.3 | 73.2 ± 0.5 | 76.6 ± 0.4 | 65.7 |
| RSC | 61.7 ± 0.8 | 53.0 ± 0.9 | 74.8 ± 0.8 | 76.3 ± 0.5 | 66.5 |
| AND-mask | 60.3 ± 0.5 | 52.3 ± 0.6 | 75.1 ± 0.2 | 76.6 ± 0.3 | 66.1 |
| SAND-mask | 59.9 ± 0.7 | 53.6 ± 0.8 | 74.3 ± 0.4 | 75.8 ± 0.5 | 65.9 |
| Fish | - | - | - | - | 66.0 |
| Fishr | 63.4 ± 0.8 | 54.2 ± 0.3 | 76.4 ± 0.3 | 78.5 ± 0.2 | 68.2 |
| CDGA-PG | 61.1 ± 1.1 | 55.9 ± 1.0 | **78.2** ± 0.8 | 79.8 ± 0.2 | 68.5 |
| CDGA*-PG | 64.0 ± 0.2 | **58.3** ± 0.4 | 77.7 ± 0.4 | **80.8** ± 0.1 | **70.2** |

Table 18: DomainBed benchmark, **DomainNet full results for training-domain validation set** model selection method.

| Algorithm | clip | info | paint | quick | real | sketch | Avg |
|---|---|---|---|---|---|---|---|
| ERM | 58.1 ± 0.3 | 18.8 ± 0.3 | 46.7 ± 0.3 | 12.2 ± 0.4 | 59.6 ± 0.1 | 49.8 ± 0.4 | 40.9 |
| IRM | 48.5 ± 2.8 | 15.0 ± 1.5 | 38.3 ± 4.3 | 10.9 ± 0.5 | 48.2 ± 5.2 | 42.3 ± 3.1 | 33.9 |
| GroupDRO | 47.2 ± 0.5 | 17.5 ± 0.4 | 33.8 ± 0.5 | 9.3 ± 0.3 | 51.6 ± 0.4 | 40.1 ± 0.6 | 33.3 |
| Mixup | 55.7 ± 0.3 | 18.5 ± 0.5 | 44.3 ± 0.5 | 12.5 ± 0.4 | 55.8 ± 0.3 | 48.2 ± 0.5 | 39.2 |
| MLDG | 59.1 ± 0.2 | 19.1 ± 0.3 | 45.8 ± 0.7 | **13.4** ± 0.3 | 59.6 ± 0.2 | 50.2 ± 0.4 | 41.2 |
| CORAL | 59.2 ± 0.1 | 19.7 ± 0.2 | 46.6 ± 0.3 | **13.4** ± 0.4 | 59.8 ± 0.2 | 50.1 ± 0.6 | 41.5 |
| MMD | 32.1 ± 13.3 | 11.0 ± 4.6 | 26.8 ± 11.3 | 8.7 ± 2.1 | 32.7 ± 13.8 | 28.9 ± 11.9 | 23.4 |
| DANN | 53.1 ± 0.2 | 18.3 ± 0.1 | 44.2 ± 0.7 | 11.8 ± 0.1 | 55.5 ± 0.4 | 46.8 ± 0.6 | 38.3 |
| CDANN | 54.6 ± 0.4 | 17.3 ± 0.1 | 43.7 ± 0.9 | 12.1 ± 0.7 | 56.2 ± 0.4 | 45.9 ± 0.5 | 38.3 |
| MTL | 57.9 ± 0.5 | 18.5 ± 0.4 | 46.0 ± 0.1 | 12.5 ± 0.1 | 59.5 ± 0.3 | 49.2 ± 0.1 | 40.6 |
| SagNet | 57.7 ± 0.3 | 19.0 ± 0.2 | 45.3 ± 0.3 | 12.7 ± 0.5 | 58.1 ± 0.5 | 48.8 ± 0.2 | 40.3 |
| ARM | 49.7 ± 0.3 | 16.3 ± 0.5 | 40.9 ± 1.1 | 9.4 ± 0.1 | 53.4 ± 0.4 | 43.5 ± 0.4 | 35.5 |
| V-REx | 47.3 ± 3.5 | 16.0 ± 1.5 | 35.8 ± 4.6 | 10.9 ± 0.3 | 49.6 ± 4.9 | 42.0 ± 3.0 | 33.6 |
| RSC | 55.0 ± 1.2 | 18.3 ± 0.5 | 44.4 ± 0.6 | 12.2 ± 0.2 | 55.7 ± 0.7 | 47.8 ± 0.9 | 38.9 |
| AND-mask | 52.3 ± 0.8 | 16.6 ± 0.3 | 41.6 ± 1.1 | 11.3 ± 0.1 | 55.8 ± 0.4 | 45.4 ± 0.9 | 37.2 |
| SAND-mask | 43.8 ± 1.3 | 14.8 ± 0.3 | 38.2 ± 0.6 | 9.0 ± 0.3 | 47.0 ± 1.1 | 39.9 ± 0.6 | 32.1 |
| Fish | - | - | - | - | - | - | 42.7 |
| Fishr | 58.2 ± 0.5 | 20.2 ± 0.2 | 47.7 ± 0.3 | 12.7 ± 0.2 | 60.3 ± 0.2 | 50.8 ± 0.1 | 41.7 |
| CDGA-PG | 61.0 ± 0.2 | 20.2 ± 0.1 | 50.7 ± 0.1 | 11.1 ± 0.3 | **65.3** ± 0.7 | **54.0** ± 0.3 | 43.7 |
| CDGA*-PG | **62.5** ± 0.0 | **24.8** ± 0.0 | **51.7** ± 0.0 | 11.7 ± 0.0 | **65.2** ± 0.0 | 52.8 ± 0.0 | **44.8** |

Table 19: DomainBed benchmark, **DomainNet full results for leave-one-out** model selection method.

| Algorithm | clip | info | paint | quick | real | sketch | Avg |
|---|---|---|---|---|---|---|---|
| ERM | 58.1 ± 0.3 | 17.8 ± 0.3 | 47.0 ± 0.3 | 12.2 ± 0.4 | 59.2 ± 0.7 | 49.5 ± 0.6 | 40.6 |
| IRM | 47.5 ± 2.7 | 15.0 ± 1.5 | 37.3 ± 5.1 | 10.9 ± 0.5 | 48.0 ± 5.4 | 42.3 ± 3.1 | 33.5 |
| GroupDRO | 47.2 ± 0.5 | 17.0 ± 0.6 | 33.8 ± 0.5 | 9.2 ± 0.4 | 51.6 ± 0.4 | 39.2 ± 1.2 | 33.0 |
| Mixup | 54.4 ± 0.6 | 18.0 ± 0.4 | 44.5 ± 0.5 | 11.5 ± 0.2 | 55.8 ± 1.1 | 46.9 ± 0.2 | 38.5 |
| MLDG | 58.3 ± 0.7 | 19.3 ± 0.2 | 45.8 ± 0.7 | 13.2 ± 0.3 | 59.4 ± 0.2 | 49.8 ± 0.3 | 41.0 |
| CORAL | 59.2 ± 0.1 | 19.5 ± 0.3 | 46.2 ± 0.1 | **13.4** ± 0.4 | 59.1 ± 0.5 | 49.5 ± 0.8 | 41.1 |
| MMD | 32.2 ± 13.3 | 11.0 ± 4.6 | 26.8 ± 11.3 | 8.7 ± 2.1 | 32.7 ± 13.8 | 28.9 ± 11.9 | 23.4 |
| DANN | 52.7 ± 0.1 | 18.0 ± 0.3 | 44.2 ± 0.7 | 11.8 ± 0.1 | 55.5 ± 0.4 | 46.8 ± 0.6 | 38.2 |
| CDANN | 53.1 ± 0.9 | 17.3 ± 0.1 | 43.7 ± 0.9 | 11.6 ± 0.6 | 56.2 ± 0.4 | 45.9 ± 0.5 | 38.0 |
| MTL | 57.3 ± 0.3 | 19.3 ± 0.2 | 45.7 ± 0.4 | 12.5 ± 0.1 | 59.3 ± 0.2 | 49.2 ± 0.1 | 40.6 |
| SagNet | 56.2 ± 0.3 | 18.9 ± 0.2 | 46.2 ± 0.5 | 12.6 ± 0.6 | 58.2 ± 0.6 | 49.1 ± 0.2 | 40.2 |
| ARM | 49.0 ± 0.7 | 15.8 ± 0.3 | 40.8 ± 1.1 | 9.4 ± 0.2 | 53.0 ± 0.4 | 43.4 ± 0.3 | 35.2 |
| VREx | 46.5 ± 4.1 | 15.6 ± 1.8 | 35.8 ± 4.6 | 10.9 ± 0.3 | 49.6 ± 4.9 | 42.0 ± 3.0 | 33.4 |
| RSC | 55.0 ± 1.2 | 18.3 ± 0.5 | 44.4 ± 0.6 | 12.2 ± 0.2 | 55.7 ± 0.7 | 47.8 ± 0.9 | 38.9 |
| CDGA-PG | 61.6 ± 0.1 | 20.6 ± 0.3 | 50.1 ± 0.4 | 11.2 ± 0.3 | **64.5** ± 0.4 | **53.8** ± 0.4 | 43.6 |
| CDGA*-PG | **62.5** ± 0.0 | **24.8** ± 0.0 | **51.7** ± 0.0 | 11.7 ± 0.0 | **65.2** ± 0.0 | 52.8 ± 0.0 | **44.8** |

Table 20: DomainBed benchmark, **DomainNet full results for test-domain validation set** (oracle) model selection method.

| Algorithm | clip | info | paint | quick | real | sketch | Avg |
|---|---|---|---|---|---|---|---|
| ERM | $58.6 \pm 0.3$ | $19.2 \pm 0.2$ | $47.0 \pm 0.3$ | $13.2 \pm 0.2$ | $59.9 \pm 0.3$ | $49.8 \pm 0.4$ | 41.3 |
| IRM | $40.4 \pm 6.6$ | $12.1 \pm 2.7$ | $31.4 \pm 5.7$ | $9.8 \pm 1.2$ | $37.7 \pm 9.0$ | $36.7 \pm 5.3$ | 28.0 |
| GroupDRO | $47.2 \pm 0.5$ | $17.5 \pm 0.4$ | $34.2 \pm 0.3$ | $9.2 \pm 0.4$ | $51.9 \pm 0.5$ | $40.1 \pm 0.6$ | 33.4 |
| Mixup | $55.6 \pm 0.1$ | $18.7 \pm 0.4$ | $45.1 \pm 0.5$ | $12.8 \pm 0.3$ | $57.6 \pm 0.5$ | $48.2 \pm 0.4$ | 39.6 |
| MLDG | $59.3 \pm 0.1$ | $19.6 \pm 0.2$ | $46.8 \pm 0.2$ | $13.4 \pm 0.2$ | $60.1 \pm 0.4$ | $50.4 \pm 0.3$ | 41.6 |
| CORAL | $59.2 \pm 0.1$ | $19.9 \pm 0.2$ | $47.4 \pm 0.2$ | $\mathbf{14.0} \pm 0.4$ | $59.8 \pm 0.2$ | $50.4 \pm 0.4$ | 41.8 |
| MMD | $32.2 \pm 13.3$ | $11.2 \pm 4.5$ | $26.8 \pm 11.3$ | $8.8 \pm 2.2$ | $32.7 \pm 13.8$ | $29.0 \pm 11.8$ | 23.5 |
| DANN | $53.1 \pm 0.2$ | $18.3 \pm 0.1$ | $44.2 \pm 0.7$ | $11.9 \pm 0.1$ | $55.5 \pm 0.4$ | $46.8 \pm 0.6$ | 38.3 |
| CDANN | $54.6 \pm 0.4$ | $17.3 \pm 0.1$ | $44.2 \pm 0.7$ | $12.8 \pm 0.2$ | $56.2 \pm 0.4$ | $45.9 \pm 0.5$ | 38.5 |
| MTL | $58.0 \pm 0.4$ | $19.2 \pm 0.2$ | $46.2 \pm 0.1$ | $12.7 \pm 0.2$ | $59.9 \pm 0.1$ | $49.0 \pm 0.0$ | 40.8 |
| SagNet | $57.7 \pm 0.3$ | $19.1 \pm 0.1$ | $46.3 \pm 0.5$ | $13.5 \pm 0.4$ | $58.9 \pm 0.4$ | $49.5 \pm 0.2$ | 40.8 |
| ARM | $49.6 \pm 0.4$ | $16.5 \pm 0.3$ | $41.5 \pm 0.8$ | $10.8 \pm 0.1$ | $53.5 \pm 0.3$ | $43.9 \pm 0.4$ | 36.0 |
| V-REx | $43.3 \pm 4.5$ | $14.1 \pm 1.8$ | $32.5 \pm 5.0$ | $9.8 \pm 1.1$ | $43.5 \pm 5.6$ | $37.7 \pm 4.5$ | 30.1 |
| RSC | $55.0 \pm 1.2$ | $18.3 \pm 0.5$ | $44.4 \pm 0.6$ | $12.5 \pm 0.1$ | $55.7 \pm 0.7$ | $47.8 \pm 0.9$ | 38.9 |
| AND-mask | $52.3 \pm 0.8$ | $17.3 \pm 0.5$ | $43.7 \pm 1.1$ | $12.3 \pm 0.4$ | $55.8 \pm 0.4$ | $46.1 \pm 0.8$ | 37.9 |
| SAND-mask | $43.8 \pm 1.3$ | $15.2 \pm 0.2$ | $38.2 \pm 0.6$ | $9.0 \pm 0.2$ | $47.1 \pm 1.1$ | $39.9 \pm 0.6$ | 32.2 |
| Fish | - | - | - | - | - | - | 43.4 |
| Fishr | $58.3 \pm 0.5$ | $20.2 \pm 0.2$ | $47.9 \pm 0.2$ | $\underline{13.6} \pm 0.3$ | $\underline{60.5} \pm 0.3$ | $50.5 \pm 0.3$ | 41.8 |
| CDGA-PG | $\underline{61.6} \pm 0.1$ | $20.9 \pm 0.2$ | $\underline{51.8} \pm 0.1$ | $12.7 \pm 0.2$ | $\mathbf{66.0} \pm 0.5$ | $\mathbf{54.4} \pm 0.2$ | $\underline{44.4}$ |
| CDGA*-PG | $\mathbf{62.5} \pm 0.0$ | $\mathbf{24.8} \pm 0.0$ | $\mathbf{51.7} \pm 0.0$ | $11.7 \pm 0.0$ | $65.2 \pm 0.0$ | $\underline{52.8} \pm 0.0$ | $\mathbf{44.8}$ |

Table 21: DomainBed benchmark, **VLCS full results for training-domain validation set** model selection method.

| Algorithm | C | L | S | V | Avg |
|---|---|---|---|---|---|
| ERM | $97.7 \pm 0.4$ | $64.3 \pm 0.9$ | $73.4 \pm 0.5$ | $74.6 \pm 1.3$ | 77.5 |
| IRM | $98.6 \pm 0.1$ | $64.9 \pm 0.9$ | $73.4 \pm 0.6$ | $\underline{77.3} \pm 0.9$ | 78.5 |
| GroupDRO | $97.3 \pm 0.3$ | $63.4 \pm 0.9$ | $69.5 \pm 0.8$ | $76.7 \pm 0.7$ | 76.7 |
| Mixup | $98.3 \pm 0.6$ | $64.8 \pm 1.0$ | $72.1 \pm 0.5$ | $74.3 \pm 0.8$ | 77.4 |
| MLDG | $97.4 \pm 0.2$ | $\mathbf{65.2} \pm 0.7$ | $71.0 \pm 1.4$ | $75.3 \pm 1.0$ | 77.2 |
| CORAL | $98.3 \pm 0.1$ | $66.1 \pm 1.2$ | $73.4 \pm 0.3$ | $\mathbf{77.5} \pm 1.2$ | $\underline{78.8}$ |
| MMD | $97.7 \pm 0.1$ | $64.0 \pm 1.1$ | $72.8 \pm 0.2$ | $75.3 \pm 3.3$ | 77.5 |
| DANN | $\mathbf{99.0} \pm 0.3$ | $\underline{65.1} \pm 1.4$ | $73.1 \pm 0.3$ | $77.2 \pm 0.6$ | 78.6 |
| CDANN | $97.1 \pm 0.3$ | $\underline{65.1} \pm 1.2$ | $70.7 \pm 0.8$ | $77.1 \pm 1.5$ | 77.5 |
| MTL | $97.8 \pm 0.4$ | $64.3 \pm 0.3$ | $71.5 \pm 0.7$ | $75.3 \pm 1.7$ | 77.2 |
| SagNet | $97.9 \pm 0.4$ | $64.5 \pm 0.5$ | $71.4 \pm 1.3$ | $\mathbf{77.5} \pm 0.5$ | 77.8 |
| ARM | $98.7 \pm 0.2$ | $63.6 \pm 0.7$ | $71.3 \pm 1.2$ | $76.7 \pm 0.6$ | 77.6 |
| V-REx | $98.4 \pm 0.3$ | $64.4 \pm 1.4$ | $\mathbf{74.1} \pm 0.4$ | $76.2 \pm 1.3$ | 78.3 |
| RSC | $97.9 \pm 0.1$ | $62.5 \pm 0.7$ | $72.3 \pm 1.2$ | $75.6 \pm 0.8$ | 77.1 |
| AND-mask | $97.8 \pm 0.4$ | $64.3 \pm 1.2$ | $\underline{73.5} \pm 0.7$ | $76.8 \pm 2.6$ | 78.1 |
| SAND-mask | $98.5 \pm 0.3$ | $63.6 \pm 0.9$ | $70.4 \pm 0.8$ | $77.1 \pm 0.8$ | 77.4 |
| Fish | - | - | - | - | 77.8 |
| Fishr | $\underline{98.9} \pm 0.3$ | $64.0 \pm 0.5$ | $71.5 \pm 0.2$ | $76.8 \pm 0.7$ | 77.8 |
| CDGA-IG | $96.3 \pm 0.7$ | $\mathbf{75.7} \pm 1.0$ | $72.8 \pm 1.3$ | $73.7 \pm 1.3$ | $\mathbf{79.6}$ |

Table 22: DomainBed benchmark, **VLCS full results for eave-one-domain-out cross-validation** model selection.

| Algorithm | C | L | S | V | Avg |
|---|---|---|---|---|---|
| ERM | 98.0 ± 0.4 | 62.6 ± 0.9 | 70.8 ± 1.9 | 77.5 ± 1.9 | 77.2 |
| IRM | **98.6** ± 0.3 | 66.0 ± 1.1 | 69.3 ± 0.9 | 71.5 ± 1.9 | 76.3 |
| GroupDRO | 98.1 ± 0.3 | 66.4 ± 0.9 | 71.0 ± 0.3 | 76.1 ± 1.4 | 77.9 |
| Mixup | 98.4 ± 0.3 | 63.4 ± 0.7 | 72.9 ± 0.8 | 76.1 ± 1.2 | 77.7 |
| MLDG | 98.5 ± 0.3 | 61.7 ± 1.2 | **73.6** ± 1.8 | 75.0 ± 0.8 | 77.2 |
| CORAL | 96.9 ± 0.9 | 65.7 ± 1.2 | 73.3 ± 0.7 | **78.7** ± 0.8 | 78.7 |
| MMD | 98.3 ± 0.1 | 65.6 ± 0.7 | 69.7 ± 1.0 | 75.7 ± 0.9 | 77.3 |
| DANN | 97.3 ± 1.3 | 63.7 ± 1.3 | 72.6 ± 1.4 | 74.2 ± 1.7 | 76.9 |
| CDANN | 97.6 ± 0.6 | 63.4 ± 0.8 | 70.5 ± 1.4 | 78.6 ± 0.5 | 77.5 |
| MTL | 97.6 ± 0.6 | 60.6 ± 1.3 | 71.0 ± 1.2 | 77.2 ± 0.7 | 76.6 |
| SagNet | 97.3 ± 0.4 | 61.6 ± 0.8 | 73.4 ± 1.9 | 77.6 ± 0.4 | 77.5 |
| ARM | 97.2 ± 0.5 | 62.7 ± 1.5 | 70.6 ± 0.6 | 75.8 ± 0.9 | 76.6 |
| VREx | 96.9 ± 0.3 | 64.8 ± 2.0 | 69.7 ± 1.8 | 75.5 ± 1.7 | 76.7 |
| RSC | 97.5 ± 0.6 | 63.1 ± 1.2 | 73.0 ± 1.3 | 76.2 ± 0.5 | 77.5 |
| ERM+GA txt2im- label | 96.5 ± 1.3 | **75.4** ± 1.4 | 71.0 ± 2.4 | 78.1 ± 1.8 | **80.3** |

Table 23: **DomainBed benchmark**, VLCS full results for test-domain validation set (oracle) model selection method.

| Algorithm | C | L | S | V | Avg |
|---|---|---|---|---|---|
| ERM | 97.6 ± 0.3 | 67.9 ± 0.7 | 70.9 ± 0.2 | 74.0 ± 0.6 | 77.6 |
| IRM | 97.3 ± 0.2 | 66.7 ± 0.1 | 71.0 ± 2.3 | 72.8 ± 0.4 | 76.9 |
| GroupDRO | 97.7 ± 0.2 | 65.9 ± 0.2 | 72.8 ± 0.8 | 73.4 ± 1.3 | 77.4 |
| Mixup | 97.8 ± 0.4 | 67.2 ± 0.4 | 71.5 ± 0.2 | 75.7 ± 0.6 | 78.1 |
| MLDG | 97.1 ± 0.5 | 66.6 ± 0.5 | 71.5 ± 0.1 | 75.0 ± 0.9 | 77.5 |
| CORAL | 97.3 ± 0.2 | 67.5 ± 0.6 | 71.6 ± 0.6 | 74.5 ± 0.0 | 77.7 |
| MMD | 98.8 ± 0.0 | 66.4 ± 0.4 | 70.8 ± 0.5 | 75.6 ± 0.4 | 77.9 |
| DANN | **99.0** ± 0.2 | 66.3 ± 1.2 | 73.4 ± 1.4 | **80.1** ± 0.5 | 79.7 |
| CDANN | 98.2 ± 0.1 | 68.8 ± 0.5 | **74.3** ± 0.6 | 78.1 ± 0.5 | 79.9 |
| MTL | 97.9 ± 0.7 | 66.1 ± 0.7 | 72.0 ± 0.4 | 74.9 ± 1.1 | 77.7 |
| SagNet | 97.4 ± 0.3 | 66.4 ± 0.4 | 71.6 ± 0.1 | 75.0 ± 0.8 | 77.6 |
| ARM | 97.6 ± 0.6 | 66.5 ± 0.3 | 72.7 ± 0.6 | 74.4 ± 0.7 | 77.8 |
| V-REx | 98.4 ± 0.2 | 66.4 ± 0.7 | 72.8 ± 0.1 | 75.0 ± 1.4 | 78.1 |
| RSC | 98.0 ± 0.4 | 67.2 ± 0.3 | 70.3 ± 1.3 | 75.6 ± 0.4 | 77.8 |
| AND-mask | 98.3 ± 0.3 | 64.5 ± 0.2 | 69.3 ± 1.3 | 73.4 ± 1.3 | 76.4 |
| SAND-mask | 97.6 ± 0.3 | 64.5 ± 0.6 | 69.7 ± 0.6 | 73.0 ± 1.2 | 76.2 |
| Fish | | | | | 77.8 |
| Fishr | 97.6 ± 0.7 | 67.3 ± 0.5 | 72.2 ± 0.9 | 75.7 ± 0.3 | 78.2 |
| CDGA-IG | 96.6 ± 0.7 | **75.5** ± 1.9 | 73.6 ± 1.1 | 77.8 ± 1.0 | **80.9** |

## G   Prompts

All prompts follow the same structure i.e., "a <class label>, <domain description>" where the domain descriptions for PACS, OfficeHome, and DomainNet are as follows:

### G.1   PACS

- Photos: photorealistic, extremely detailed
- Sketches: sketch drawing, black and white, less details
- Cartoons: cartoon, cartoonish
- Art: art painting

### G.2   OfficeHome

- Clipart: Clipart, schematic, simplified
- Product: Product, Merchandise
- Real: Real World, extremely detailed
- Art: art painting, art

### G.3   Domainnet

- Clipart: cartoon, cartoonish, drawing
- Infograph: infographic, data visualization, poster
- Real: photorealistic, extremely detailed
- Painting: art painting
- Quickdraw: extremely simple drawing, black and white
- Sketch: sketch drawing, black and white, less details
- Clipart: cartoon, cartoonish, drawing

## H   Code

To reproduce the DomainBed results, each class-specific dataset object inherits from either CDGA or CDGA* classes provided in this section. See the script provided in the section H.

```
1  class CDGA(MultipleDomainDataset):
2      def __init__(self, root, test_envs, augment, hparams):
3          super().__init__()
4
5          transform = transforms.Compose(
6              [
7                  transforms.Resize((224, 224)),
8                  transforms.ToTensor(),
9                  transforms.Normalize(
10                     mean=[0.485, 0.456, 0.406], std=[0.229, 0.224, 0.225]
11                 ),
12             ]
13         )
14
15         augment_transform = transforms.Compose(
16             [
17                 # transforms.Resize((224,224)),
```

```
18                  transforms.RandomResizedCrop(224, scale=(0.7, 1.0)),
19                  transforms.RandomHorizontalFlip(),
20                  transforms.ColorJitter(0.3, 0.3, 0.3, 0.3),
21                  transforms.RandomGrayscale(),
22                  transforms.ToTensor(),
23                  transforms.Normalize(
24                      mean=[0.485, 0.456, 0.406], std=[0.229, 0.224, 0.225]
25                  ),
26              ]
27          )
28
29          environments = [f.name for f in os.scandir(root) if f.is_dir()]
30          environments = sorted(environments)
31
32          self.datasets = []
33          print(f"Test domains: {test_envs}")
34          for i, environment in enumerate(environments):
35              # Transformation
36              if augment and (i not in test_envs):
37                  env_transform = augment_transform
38              else:
39                  env_transform = transform
40
41              path = os.path.join(root, environment)
42              # Create list of generated subfolders for each distribution
43              sub_environments = [f.name for f in os.scandir(path) if f.is_dir()]
44              if i not in test_envs:
45                  # if we are in the training distribution combine folders that are not in the
       test distributions
46                  env_dataset = []
47                  for sub_env in sub_environments:
48                      if all(environments[i] not in sub_env for i in test_envs):
49                          print(f"Adding {sub_env} to {environment} for training")
50                          env_dataset.append(
51                              ImageFolder(
52                                  os.path.join(path, sub_env), transform=env_transform
53                              )
54                          )
55                  self.datasets.append(torch.utils.data.ConcatDataset(env_dataset))
56              else:
57                  # if we are in the testing distribution just use the original data
58                  print(f"using {environment} for testing")
59                  self.datasets.append(
60                      ImageFolder(
61                          os.path.join(path, environment), transform=env_transform
62                      )
63                  )
64          self.input_shape = (
65              3,
66              224,
67              224,
68          )
69
70
71  class CDGA_star(MultipleDomainDataset):
72      def __init__(self, root, test_envs, augment, hparams):
73          super().__init__()
74
75          transform = transforms.Compose(
76              [
77                  transforms.Resize((224, 224)),
78                  transforms.ToTensor(),
79                  transforms.Normalize(
80                      mean=[0.485, 0.456, 0.406], std=[0.229, 0.224, 0.225]
81                  ),
82              ]
83          )
84
```

```
85          augment_transform = transforms.Compose(
86              [
87                  # transforms.Resize((224,224)),
88                  transforms.RandomResizedCrop(224, scale=(0.7, 1.0)),
89                  transforms.RandomHorizontalFlip(),
90                  transforms.ColorJitter(0.3, 0.3, 0.3, 0.3),
91                  transforms.RandomGrayscale(),
92                  transforms.ToTensor(),
93                  transforms.Normalize(
94                      mean=[0.485, 0.456, 0.406], std=[0.229, 0.224, 0.225]
95                  ),
96              ]
97          )
98
99          environments = [f.name for f in os.scandir(root) if f.is_dir()]
100          environments = sorted(environments)
101
102          self.datasets = []
103          print(f"Test domains: {test_envs}")
104          for i, environment in enumerate(environments):
105              if augment and (i not in test_envs):
106                  env_transform = augment_transform
107              else:
108                  env_transform = transform
109              path = os.path.join(root, environment)
110              # create list of generated subfolders for each distribution
111              sub_environments = [f.name for f in os.scandir(path) if f.is_dir()]
112              if i not in test_envs:
113                  # if we are in the training distribution combine all the test folder except
    the original test data
114                  env_dataset = []
115                  for sub_env in sub_environments:
116                      print(f"Adding {sub_env} to {environment} for training")
117                      env_dataset.append(
118                          ImageFolder(
119                              os.path.join(path, sub_env), transform=env_transform
120                          )
121                      )
122                  self.datasets.append(torch.utils.data.ConcatDataset(env_dataset))
123              else:
124                  # if we are in the testing distribution just use the original data
125                  print(f"using {environment} for testing")
126                  self.datasets.append(
127                      ImageFolder(
128                          os.path.join(path, environment), transform=env_transform
129                      )
130                  )
131          self.input_shape = (
132              3,
133              224,
134              224,
135          )
136
137
138  class G_PACS(CDGA):
139      CHECKPOINT_FREQ = 300
140      ENVIRONMENTS = ["A", "C", "P", "S"]
141      num_classes = 7
142
143      def __init__(self, root, test_envs, hparams):
144          self.dir = os.path.join(root, "G_PACS/")
145          super().__init__(self.dir, test_envs, hparams["data_augmentation"], hparams)
```