# OpenReview forum: "Beyond Loss Functions: Exploring Data-Centric Approaches with Diffusion Model for Domain Generalization"
_TMLR — Accepted by TMLR_

### Review · Reviewer_tdhq · 2024-08-29

**Summary Of Contributions:**

The authors propose a method for learning good cross-domain classifiers by using diffusion models to adapt data across domains. With the use of strong diffusion models like Stable Diffusion they are able to achieve state-of-art results on benchmarks like DomainBed. They also attempt to explain why their method works with the vicinal risk minimization principle and provide detailed empirical analysis on their method reduces domain shifts.

**Audience:**

Yes

**Claims And Evidence:**

Yes

**Requested Changes:**

- Improve the clarity by rewriting Section 3 and moving some of the important information from the Appendix to the main paper

- Explain how we can ensure the benefit of the CDGA method comes from cross-domain transfer and not from generating extra synthetic data from trained diffusion models like Stable Diffusion.

**Strengths And Weaknesses:**

Strengths:
- The proposed approach is effective. There are significant improvements on the DomainBed benchmark against other state-of-art methods for cross-domain generalization.

- The authors attempt to explain the effectiveness of their method by the vicinal risk minimization principle by Chapelle et al 2000.

- There is a fairly comprehensive set of experiments to measure how the CDGA method reduces domain shift across datasets.

- The authors also provide how the models trained with CDGA reduces loss landscape sharpness and improves adversarial robustness


Weaknesses:
- I have reservations with the use of strong models like Stable Diffusion trained on a lot of image data for data transformation across domains. The main problem is we cannot tell whether the benefit comes from generalizing from cross-domain data, or just by having a strong diffusion model that can generate extra examples for learning in the target domain. For example in Figure 5, the generated dog images do not look like the same type of dogs as the images used as guides. It is difficult to interpret the model as performing cartoon -> cartoon or cartoon -> sketch transformation as the source and target have little in common apart from being images of dogs. A control experiment can be done by generating images in the target domain (sketch, cartoon) by providing the word prompt 'dog' only and see how the accuracy improves by training on these generated data, without using any guides from other domains. In this way we can more accurately measure the effect of the source of improvement.

- Some of the information presented in the appendix should be included in the main paper. For example, the first page of Appendix A includes important information about the method and should be included in the main paper. Also, Figure 10 comparing CDGA and SDGA is very informative and should be included too.

- Section 3 is not clearly written. Shouldn't the number of examples for domain S_i depend on the number of examples of other domain S_j as the extra examples are translated from other domains? To clarify the authors can just write down the ERM equation like Equation 2. Also the definition of CDGA* is unclear. Is it equivalent to CDGA-PG or CDGA-IG?

---

> ### Author Response · Authors · 2024-09-16
> **Addressing Requested changes by Reviewer tdhq**
>
> We thank the reviewer for the positive and insightful comments. Here, we answer each point raised by the reviewer and update the manuscript whenever applicable.
>
>
> - Improve the clarity by rewriting Section 3 and moving some of the important information from the Appendix to the main paper: We moved the first page of Appendix A to the main paper. For Figure 10 which compares CDGA and SDGA, if there is any space left, we attempt to include it in the main manuscript.
>
> ---
>
> - Section 3 is not clearly written. Shouldn't the number of examples for domain S_i depend on the number of examples of other domain S_j as the extra examples are translated from other domains? We polish section 3 of the paper to make it clearer. Regarding the number of examples for domain $S_i$, note that for each example from domain $S_i$, we map this example to all other possible domains using guidances from those domains. As a result, if there are $|S_i|$ examples for the original domain $S_i$, after augmentation, the number of
>  generated samples is  $|S_i| × (b × n)$ where $b$ is the generation batch size (number of samples generated from LDM after each call given an image and guidance)  and $n$ is the number of domains. Finally, because we also include the original data with the synthetic data for training, the number of training samples will be  $|S_i| × (b × n) +|S_i|$ =   $|S_i| * (b × n + 1)$.
>
> ---
>
> - The definition of CDGA* is unclear. Is it equivalent to CDGA-PG or CDGA-IG? CDGA means there are guidance attributes all training domains while CDGA* means the guidance attributes can also include guidance from the target domain. With these explanations, if the guidance attribute is an image, we call the method CDGA-PG, and if the guidance is textual, the method is named CDGA-IG. So all in all, both CDGA-PG and CDGA-IG can become CDGA*-PG and CDGA*-IG respectively if the guidance also includes the target domain information. We think this confusion comes from our naming in tables in which we used CDGA-PG*. To resolve the confusion, we replace  CDGA-PG*  with  CDGA*-PG in all tables.
>
>
> ---
>
> -  Explain benefit of the CDGA method comes from cross-domain transfer and not from generating extra synthetic data from trained diffusion models like Stable Diffusion: We thank the reviewer for raising this point. We understand the reviewer's concern and during conducting this research we also thought about this point. We do believe the superior OOD performance of CDGA is due to cross-domain transfer which removes the shift between domains and not just because we are using a model that is pretrained and generates synthetic data. The reason behind this is the experimental results from the domainbed benchmark where a large imagenet pretrained model (ResNet) is used for all algorithms in Tables 1-4 in our paper. As you can see, even with a large pretrained model, the model fails to achieve good OOD generalization while employing CDGA further boost the performance. We add a discussion about this point in the paper as other readers may raise this question.

---

### Review · Reviewer_xJm6 · 2024-09-05

**Summary Of Contributions:**

This paper employs diffusion models to address the problem of domain generalization. In the proposed CDGA method, cross-domain synthetic images are generated using diffusion models and serve as additional training data to vanilla ERM. This method is formulated as a type of cross-domain VRM (Vicinal Risk Minimization) in the analysis. In the experiments, authors show the CDGA variants outperform several SOTA domain generalization algorithms on the benchmarking platform DomainBed. Authors also show through several measurements that the cross-domain images reduce the differences between domains.

**Audience:**

Yes

**Claims And Evidence:**

No

**Requested Changes:**

**Critical**:
1. In the experiment results on domain generalization performance (Table 1~4), the best-performing method is often CDGA*, which assumes access to guidance attributes from the target domain. Do other baseline (SOTA DG) methods considered in these results also have access to information from target domain? My main concern is whether the comparison is fair in terms of access to the target domain.
2. How does the number of training samples in the proposed CDGA method compare to that of the other SOTA methods?
3. What is the distance measurement $d(\cdot, \cdot)$ in the domain projection operator when instantiated with a latent diffusion model? I did not find further details of this in section 5.
4. What are the assumptions needed for the kernel function, in theoretical claims about VRM? For example, does the kernel need to satisfy positivity, and/ or certain smoothness assumption, and/or translation-invariance? I didn’t see these conditions specified.
5. Closely following point 4, I am confused about the kernel induced from CDGA. Without any theoretical evidence that CDGA in fact induces a kernel K which satisfies the assumptions needed in VRM (as specified in the expression above equation (8) on page 7), I don’t think that the claims of theoretical equivalency between CDGA and ERM(VRM) with a different kernel is well-supported. Due to the concerns with this part, I am leaving the answer to "Claims And Evidence" as "No" for now.
6. In the expression above equation (8) on page 7, it states that $x$ is drawn from $K (\cdot; P_{i\rightarrow j}(x_k^i))$. However, $K (\cdot; P_{i\rightarrow j}(x_k^i))$ is a function, not a data distribution. If the authors meant to define a probability distribution from the kernel function (e.g. by normalizing), they should make it mathematically correct.
7. In Section 8, regarding the results on adversarial robustness, I am interested to know how the baseline methods (considered in Table 1~4) perform in terms of adversarial robustness.

**Nice to have**:
1. Why is it that for other datasets than VLCS, the guidance attributes are text prompts, but for VLCS the attributes are images?

**Strengths And Weaknesses:**

I am not very familiar with the fields of domain generalization and diffusions models, so my review should definitely be taken with a grain of salt when it comes to these aspects.

**Strengths**:
1. The paper is clearly structured and mostly well-written.
2. The idea of using diffusion models to generate cross-domain images to address domain generalization is novel, unless there are any prior works that I have overlooked.

**Weakness**:
1. There are some concerns about the comparison between the proposed method and SOTA methods. See requested changes.
2. The theoretical analysis contains many parts that are ambiguous / not rigorous. I would suggest authors make the statements about connection to VRM more rigorous and scale back claims where necessary. For further details, please see requested changes 3~6.

---

> ### Author Response · Authors · 2024-09-19
> **Addressing Requested changes by Reviewer xJm6**
>
> We thank the reviewer for the comprehensive, and insightful comments. Here, we answer each point raised by the reviewer and update the manuscript accordingly.
>
> - Results on domain generalization performance (Table 1~4):
>
> First note that CDGA that only uses training domain guidance attributes also outperforms the SOTA  DG methods and is the second-best performing method after CDGA*. With this in mind, CDGA achieves the best DG performance without target domain guidance compared with other baselines. CDGA* is a possible solution for further OOD generalization if we have a textual description of what the target domain may look like. Such information is not useful for other algorithms as there is no mechanism to exploit it. Further, In practice, this textual information can be available in many applications. For example,  we may know the pictures in the target domain describe a dry area that can be the guidance.
>
> ---
>
> - How does the number of training samples in the proposed CDGA method compare to that of the other SOTA methods?
>
> For CDGA, the number of augmented examples for domain $S_i$ is  $|S_i| × (b × n)$ where $b$ is the generation batch size (number of samples generated from LDM after each call given an image and guidance)  and $n$ is the number of training domains. As a result, because we also include the original data with the synthetic data for training, the number of training samples for domain $S_i$ in CDGA will be  $|S_i| × (b × n) +|S_i|$ =   $|S_i| * (b × n + 1)$. With similar reasoning, the number of training examples for domain $S_i$ in CDGA* is $(b \times (n+1) + 1) \times | S_i |$. For most of the other baselines, the number of examples for domain $S_i$ is $|S_i $|.
>
> ---
>
> - The distance measurement in the domain projection operator:
>
> According to Vapnik (1999) (The Nature of Statistical Learning Theory (Second Edition), page 269), distance can be a metric in space to define vicinity. For example, one can use $L_1$, $L_2$, or $l_{\infty}$ metrics to define the vicinity neighborhood.  For the domain projection operator, we follow a similar path where the only difference is that the vicinity defined is between domains. We add more details about this in the manuscript to make it clear.
>
>
> ---
> - What are the assumptions needed for the kernel function, in theoretical claims about VRM?
>
> According to Chapelle et al. 2000 and Vapnik 1999, any kernel that can approximate the local density can be an option and the pointwise kernel estimates are not the only way to go. Smoothness is not necessarily required. However,  according to Vapnik 1999, the common assumption is that the true data distribution (which is unknown) is smooth. As a result, this motivates us to propose smooth kernels for density estimates as alternatives to non-smooth pointwise estimates offered by ERM for better density estimates.
>
>
> - How CDGA  induces a kernel K which satisfies the assumptions needed in VRM:
>
> The fact that using augmented data from vicinal distribution can act as a kernel has already been shown in the literature. In particular, it is easy to show that using the augmented data in the Gaussian vicinity is equivalent to augmenting the training data with additive Gaussian noise. As another example, Mixup by Hongyi Zhang (2018) proposed mixup data augmentation as a technique for improving true data distribution. Mixup vicinal distribution is constructed by a convex combination of two images and their labels.
>
> We are following the same idea where the augmented data from the vicinal distributions between domains are generated using the LDM   given data from the source domain and guidance from the target domain. In the paper, we are NOT making any theoretical claims that  using synthetic data from LDM is equivalent to a kernel that improves better estimation of true data distribution. Even Chapelle et al. (2000) do not provide any theoretical conditions on the kernel to provide a better estimation of true data distribution. The only intuition is kernel leads to additional data points in the vicinity of training samples can improve the estimation error. However, for LDM, because the model is pre-trained on a large dataset and has great data manipulation ability, we expect the LDM to generate additional synthetic data in the vicinity of domains that act as a kernel that hopefully improves the true data distribution
> and $x\mapsto \mathbb E(M(x,guide^j))$ approximates the projection operator $P_{i\rightarrow j}$
>
>
> ---
> - $K(\cdot,P_{i\to j})$ is function or distribution:
> Thanks for mentioning this point. K(\cdot,P_{i\to j})$  refers to function which is realized by sampling from LDM.  We modify the text accordingly to make this clear.
> ---
> Why is it that for other datasets than VLCS, the guidance attributes are text prompts, but for VLCS the attributes are images?
> This is due to the nature of VLCS where domains cannot be easily described by textual prompts. In VLCS, domains are Caltech101, LabelMe, SUN09, and VOC2007 classes.

---

> > ### Comment · Reviewer_xJm6 · 2024-10-12
> > **Reply to author's response**
> >
> > Apologies for my late response, and thanks for the clarifications.
> >
> > While I understand that authors are not claiming that CDGA achieves better estimation of the true data distribution, I am still confused about the claim that CDGA is equivalent to replacing pointwise kernel estimates in ERM with new density estimates. In the simpler case, augmenting training data with random Gaussian noise is equivalent to drawing data from a distribution that is smoothed from the delta distribution (at data point x_k) with a Gaussian kernel. However, for CDGA, if it indeed generates data from a distribution smoothed from the delta distribution at $P_{i\rightarrow j}(x_k^j)$ (as stated in the paper above equation (8): $\tilde{x_k}^{ij} \sim K(\cdot;P_{i\rightarrow j}(x_k^j))$, I am unclear on *what* exactly the kernel K is in this case. Additionally, I don’t see the evidence that CDGA (domain $i\rightarrow j$) generates data in domain j that is around the datapoint *closest* to domain i… To summarize, I don’t see how $\tilde{x_k}^{ij} \sim K(\cdot;P_{i\rightarrow j}(x_k^j))$ is justified, without any specifications on K, and more generally the relationship between $\tilde{x_k}^{ij}$ and $P_{i\rightarrow j}(x_k^j)$.

---

> > > ### Author Response · Authors · 2024-10-15
> > > **Reply to Reviewer xJm6**
> > >
> > > Considering that $K(\cdot;P_{i\to j}(x_{k}^{i}))$ is a kernel in our definition, we corrected the error in the equation above equation 8, by modifying it from
> > >
> > > $x_{k}^{i,j} = \textit{M}(x_{k}^{i},\texttt{guide}^{j}) \sim K(\cdot;P_{i\to j}(x_{k}^{i})).$
> > >
> > >  to:
> > >
> > > $x_{k}^{i,j} = K(\cdot;P_{i\to j}(x_{k}^{i}))  \sim M(x_{k}^{i},\texttt{guide}^{j}).$
> > >
> > > Note that the kernel $K(\cdot;P_{i\to j}(x_{k}^{i}))$  is realized by sampling from LDM. So regarding the reviewer's question "what exactly the kernel K is in this case?" The answer is that we are not specifying any explicit form of function for this kernel in our paper, instead, we suggest that such kernel $K(\cdot;P_{i\to j}(x_{k}^{i}))$ can be realized using a diffusion model given an image in domain $i$ and a guidance attribute from domain $j$. The suggestion that kernel $K(\cdot;P_{i\to j}(x_{k}^{i}))$ can be realized using a diffusion model or equivalently the evidence that CDGA (domain $i\to j$) generates data in domain $j$ that is around the datapoint closest to domain $i$ is NOT based on any theoretical proof. However, the proven superior image manipulation ability of the diffusion models for mixing images sampled from different distributions in the literature and more importantly our experimental results in reducing domain shift after adding augmented data to the real data in Figure 2 and Figure 6 (left) support this claim.
> > >
> > > To be more clear, if the reviewer considers t-SNE plot in Figure 2, for domain $A$ and $S$ for the real data, it can be seen from the t-SNE plot that the synthetic samples $A \to S$ exist in the manifold of meaningful data points between the two domains $A$ and $S$.
> > >
> > > We make this clearer by updating the manuscript where the realization of the kernel function is discussed. We add more details on reasons why we think sampling from LDM can realize such a kernel in practice.

---

> > > > ### Comment · Reviewer_xJm6 · 2024-10-22
> > > > **Additional reply to author's response**
> > > >
> > > > Thank you for the quick reply. I did not see the updated manuscript -- please let me know when you upload the updated version so I can look at the corresponding passages.
> > > >
> > > > Re kernels: I believe that the form is still not correct, and more importantly, the claim that sampling from LDM is equivalent to sampling from the density defined by $K(\cdot; P_{i\rightarrow j}(x_k^i))$ is still not correct. Let us revisit what the claims are -- am I correct in that (I) authors want to define a distribution that is (a) centered around $P_{i\rightarrow j}(x_k^i)$; (b) is smoothed by some kernel $K(\cdot)$; and (II) authors are suggesting that sampling from the data generating distribution of the LDM is equivalent to sampling from that distribution?
> > > >
> > > > For (I), note that the probability density should be defined as $p(\cdot) = \frac{K(\cdot; P_{i\rightarrow j}(x_k^i)))}{Z}$ where Z is the normalizing constant $\int K(x; P_{i\rightarrow j}(x_k^i))) dx$. So the correct way to express that some variable $X$ is drawn from this distribution is $X\sim P$ where $p(x)$ is as defined.
> > > >
> > > > For (II), I don't see how the empirical evidence in Figure 2 support the claim that $x_k^{i,j} = M(x_k^i, \text{guide}^j)\sim P$ with $p(x)$ defined above. I agree that Figure 2 suggest that $M(x_k^i, \text{guide}^j)$ generates random data points between the two domains $i (A)$ and $j (S)$.  This data generating distribution, however, is NOT centered at $P_{i\rightarrow j}(x_k^i)$, which itself should be a point *inside* domain $j (S)$ by definition.
> > > >
> > > > If I didn't misunderstand the above, I suggest the authors to scale back claims such as " From a theoretical standpoint, we illustrate that CDGA is equivalent to replacing pointwise kernel estimates in ERM with new density estimates in the proximity of domain pairs" -- because the equivalency is unsupported, and there lacks theoretical results. And to please rewrite claims in Section 5 (such as the expression above equation (8)) to accurately reflect what is going on.

---

> > > > > ### Author Response · Authors · 2024-10-24
> > > > > **Additional response to Reviewer xJm6**
> > > > >
> > > > > We thank the reviewer for the constructive comments.
> > > > >
> > > > > Answer to point (I): Yes. we aim to define a distribution that is centered around $P_{j}(x_{k}^{i})$ and is smoothed by some kernel $K(\cdot)$. We modified this definition by adding the normalizing constant to the definition in the updated manuscript.
> > > > >
> > > > >
> > > > > Answer to point (II):
> > > > > Yes. we are suggesting that sampling from the data generating distribution of the LDM while it projects data from domain $i$ to domain $j$ can be seen as sampling from that distribution $K(x;P_{j}(x_{k}^{i}))$. We do agree with the reviewer this claim is not theoretically proven.
> > > > >
> > > > > Considering this, we removed all statements that mention theoretically, sampling from the data-generating distribution of the LDM is equivalent to sampling from the $\frac{K(\cdot;P_{i\to j}(x_{k}^{i}))}{\int K(\cdot;P_{i\to j}(x_{k}^{i}))\, dx}$ in the updated manuscript. Instead, we consider this as an intuitive justification that explores the connection of CDGA with VRM and possible reasons why CDGA with ERM improve the DG performance of ERM.

---

### Review · Reviewer_Azv7 · 2024-09-05

**Summary Of Contributions:**

The paper introduces a data-centric approach that integrates cross-domain data augmentation (CDGA) with the empirical risk minimization (ERM) algorithm, achieving improved state-of-the-art (SOTA) performance in terms of accuracy on both training and test datasets. CDGA leverages latent diffusion models to perform transformations that combine an input from one domain with a guidance attribute from a different domain, but within the same class. The guidance attribute can either be a prompt or an image, with the two variants including prompt-based guidance (CDGA-PG), which utilizes class labels or descriptions, and image-based guidance (CDGA-IG).

The authors demonstrate several key properties of the CDGA + ERM framework: (1) it generalizes the vicinal risk minimization (VRM) principle to the domain generalization setting, (2) t-SNE analysis shows that it effectively reduces domain shift, (3) it promotes the formation of flatter local minima during training, and (4) it enhances adversarial robustness. The training and test data are sourced from DomainBed, including datasets such as PACS, Office-Home, DomainNet, and VLCS. The CDGA approach consistently outperformed baseline methods, particularly outperforming the standard ERM algorithm.

Additionally, ablation studies were conducted to assess the performance of CDGA against single-domain generative augmentation (SDGA) and with varying data sizes. The results indicate that CDGA shows significant potential compared to its SDGA counterparts.

**Audience:**

Yes

**Broader Impact Concerns:**

The work in its current form does not exhibit any broader impact concerns.

**Claims And Evidence:**

Yes

**Requested Changes:**

It would be beneficial to move some of the results currently in the appendix to the main paper, while shifting the sections on loss function landscape analysis and adversarial robustness to the appendix. For example, specific examples of the prompts or images used as guidance attributes could be included in the main paper to enhance clarity. As mentioned in the weaknesses, adding a section on the statistical significance of the performance differences between the algorithms would also be valuable. Additionally, the section on mitigating class imbalance could be moved to the main paper. Lastly, offering deeper insights into the comparison between SDGA and CDGA would be helpful, as the results suggest that SDGA performs only slightly worse than CDGA. Highlighting scenarios in which CDGA is preferable over SDGA would provide useful context for readers.

**Strengths And Weaknesses:**

**Strengths:**
- The proposed cross-domain data augmentation (CDGA) for dataset augmentation is a novel and interesting approach. The domain shift analysis of CDGA is particularly insightful, as it effectively demonstrates the advantages of the algorithm over conventional ERM methods.
- The evaluation is thorough, utilizing four different benchmark datasets and comprehensive comparisons with state-of-the-art (SOTA) algorithms.
- The ablation study involving single-domain generative augmentation (SDGA) offers valuable insights and opens up interesting avenues for future research. These results, in particular, could benefit from further theoretical exploration.
- Overall, the paper is well-written and clear in its presentation.

**Weakness:**
- While the main contributions of the paper are well-justified, the inclusion of the loss landscape analysis and adversarial robustness study feels somewhat tangential to the core findings. These sections could be relocated to the appendix for better focus.
- A deeper analysis of the ablation studies would have been more beneficial, offering additional insights into the functioning of the proposed augmentation technique.
- Finally, incorporating statistical significance tests would further strengthen the results. For example, in the ablation studies, was the performance difference between CDGA and SDGA statistically significant? Similarly, in the DomainBed benchmarks, did ERM + CDGA significantly outperform the MixUp method? Statistical tests would provide clearer evidence of the method's advantages.

---

> ### Author Response · Authors · 2024-09-15
> **Addressing Requested changes by Reviewer Azv7**
>
> We thank you for your positive and insightful comments. Following your requested changes, we answer each point separately and modify the manuscript.
>
> -  Loss function landscape analysis and adversarial robustness sections: Following the reviewer's suggestion, we shifted the sections on loss function landscape analysis and adversarial robustness to the appendix.
> ---
> -  Adding examples of the prompts or images used as guidance attributes: To enhance clarity, we included specific examples of the prompts or images used as guidance attributes in the main paper.
> ---
> -  Regarding the statistical significance of the performance differences: For the results on the performance of each algorithm, we followed the scheme suggested by the domainbed benchmark which is the most common and standard way to compare DG methods in a fair way. Please consider every accuracy number reported here given an algorithm and a dataset is the results of 3 independent trials where each was run for 20 random hyperparameter choices where 3 different model selection techniques have been used to select the best hyperparameters and results are averaged over trials. This suggests that the numbers reported here are robust and meaningful and not due to any noise training process or selection of hyperparameters.
> ---
> - Mitigating class imbalance: Following the reviewer's suggestion, we moved the mitigating class imbalance section from the appendix to the main paper.
> ---
> - Deeper insights into the comparison between SDGA and CDGA: We believe as long as the objective is maximum OOD performance, we have access to either textual or visual description of different domains and there is no computational bottleneck, the CDGA is a better choice compared with SDGA. On the other hand, if we do not have access to the domain descriptions, or we want to get OOD improvement as fast as possible with fewer training examples, SDGA can be a better option. We included additional discussion regarding the comparison of SGA and CDGA in the paper.

---

> > ### Comment · Reviewer_Azv7 · 2024-09-25
> >
> > Thank you authors for implementing the requested changes and clarifying the comparison between SDGA and CDGA!

---

### Decision · Action_Editor_v3ni · 2024-11-04

**Recommendation:** Accept with minor revision

**Comment:**

The recommendation is based on the reviewers' comments, the action editor's evaluation, and the authors’ response.

This paper studies the use of pretrained diffusion models as a data augmentation generator to facilitate domain generalization. All reviewers find the studied setting novel and the results provide new insights. The authors’ rebuttal has successfully addressed the major concerns of reviewers. Therefore, I recommend acceptance of this submission.

However, the authors are suggested to add more clarity on explaining the source of performance gain when using stable diffusion models, given that these models are trained on large-scale image data, and it is not clear whether the considered tasks are considered "out-of-domain" relative to the pertaining data. Arguing ResNet pretrained on ImageNet fail in domain transfer does not fully justify this point, because the pertaining data of stable diffusion are much more diverse and more in volume compared to ImageNet.

I also expect the authors to include the new results and suggested changes during the rebuttal phase to the final version.

**Audience:**

The data augmentation setup with generative models is of broad interest to the machine learning community.

**Claims And Evidence:**

The paper is clearly written. The hypotheses are well articulated and supported by the empirical results.

---

> ### Author Response · Authors · 2024-11-14
> **Reply to Action Editor v3ni**
>
> We appreciate your positive and insightful comments.
>
> We addressed all the comments by the reviewers in the final version of the paper. These changes include:
>
> 1. We removed al the statements in the paper that claim theoretically sampling from LDM is equivalent to sampling from the density defined by $K(\cdot;P_{i\to j}(x_{k}^{i}))$.
>
> 2.  We modified the equation above equation 8 by adding the normalizing constant.
>
> 3. We shifted the sections on loss function landscape analysis and adversarial robustness to the appendix.
>
> 4. We moved the mitigating class imbalance section from the appendix to the main paper.
>
> 5. We added a discussion on the comparison between SDGA and CDGA. This part considers the advantage of each method compared to the other one and can be found on page 11 (section 7.2).
>
> 6.  We moved the first page of Appendix A to the main paper.
>
> Specially, regarding the source of OOD generalization performance gain, we added section 8 on page 11 and 12 to justify our claim that superior OOD generalization is due to the cross-domain transfer ability of CDGA and not just because this LDM is trained on large-scale image dataset. In the following you can see this section.
>
>
>
> #  Does superior performance of the CDGA method comes from cross-domain transfer or the improvement is just because of extra synthetic images from a pre-trained diffusion model
> A critical question that may be raised is whether the benefit of the CDGA method comes from cross-domain transfer and not from generating extra synthetic data from a pre-trained diffusion model. We do believe the superior OOD performance of CDGA is due to cross-domain transfer which removes the distribution shift between domains and not just because we are using a model that is pre-trained and generates synthetic data. Here we provide our arguments to support this claim.
>
>
>
> *  **The reason behind beter OOD generalization is synthetic samples *between domains (cross domain augmentation)* and not just additional samples from a pre-trained diffusion model**. By looking more closely at Table. 6, we observe that although in both SDGA and CDGA techniques the same pre-trained diffusion model has been used, the OOD generalization improvement for CDGA is much higher than the improvement by SDGA. This observation suggests that the superior OOD generalization is not just due to incorporating additional syntethic samples. Instead, this improvement is due to employing **additional syntethic samples that reduce the distribution shift across domains**. The claim that CDGA actually reduces the distribution shift more than SDGA is experimentally supported in Figure 4  where for example A $\rightarrow C$ synthetic samples from CDGA are filling the gap between domains A and C while for SDGA, only A $\rightarrow A$ samples are generated which are not able to reduce the domain shift between domain A and C. More examples on this can be found in Figure 15 in the supplementary material. In other words, while both SDGA and CDGA employ the same pre-trained model, CDGA is able to generate samples between domains and reduce the distribution shift better than SDGA and as a result achieve better OOD generalization.
>
> * **We are already employing pre-trained ResNet for the domainbed experiments and it fails to provide good OOD generalization**. Consider the experimental results from the domainbed benchmark where a large imageNet pre-trained model (ResNet) is used for all algorithms in Tables 1-4 in our paper. As you can see, even with a large pre-trained model, the model fails to achieve good OOD generalization while employing CDGA further boosts the performance.